# Overcoming optical losses in thin metal-based recombination layers for efficient n-i-p perovskite-organic tandem solar cells

Jingjing Tian [1,2], Chao Liu [1,3] ✉, Karen Forberich [3], Anastasia Barabash[1], Zhiqiang Xie[1], Shudi Qiu[1], Jiwon Byun [4], Zijian Peng[1,2], Kaicheng Zhang [1], Tian Du[3], Sanjayan Sathasivam[5], Thomas J. Macdonald[6], Lirong Dong[1,2], Chaohui Li [1,2], Jiyun Zhang [3], Marcus Halik[4], Vincent M. Le Corre [3], Andres Osvet[1], Thomas Heumüller [1,3], Ning Li [7], Yinhua Zhou [8], Larry Lüer [1] ✉ & Christoph J. Brabec [1,3] ✉

Perovskite-organic tandem solar cells (P-O-TSCs) hold substantial potential to surpass the theoretical efficiency limits of single-junction solar cells. However, their performance is hampered by non-ideal interconnection layers (ICLs). Especially in n-i-p configurations, the incorporation of metal nanoparticles negatively introduces serious parasitic absorption, which alleviates photon utilization in organic rear cell and decisively constrains the maximum photocurrent matching with front cell. Here, we demonstrate an efficient strategy to mitigate optical losses in Au-embedded ICLs by tailoring the shape and size distribution of Au nanoparticles via manipulating the underlying surface property. Achieving fewer, smaller, and more uniformly spherical Au nanoparticles significantly minimizes localized surface plasmon resonance absorption, while maintaining efficient electron-hole recombination within ICLs. Consequently, optimized P-O-TSCs combining $CsPbI_2Br$ with various organic cells benefit from a substantial current gain of >1.5 mA/cm² in organic rear cells, achieving a champion efficiency of 25.34%. Meanwhile, optimized ICLs contribute to improved long-term device stability.

Tandem photovoltaics with complementary absorbing sub-cells are a promising pathway to surpass the detailed balance limit of single-junction solar cells, stimulating intense interest[1,2]. Over the past decade, metal-halide perovskites (PVKs) have experienced a significant upswing as light absorbers in the photovoltaics community due to

their unique optoelectronic properties[3,4], leading to a certified power conversion efficiency (PCE) of 26.7% for single-junction perovskite solar cells (PSCs)[5]. Essentially, their flexible bandgap tunability (1.17–3.10 eV) renders PSCs an ideal choice as both front and rear sub-cells in tandem solar cells (TSCs)[2]. In addition, the emerging organic

[1]Institute of Materials for Electronics and Energy Technology (i-MEET), Department of Materials Science and Engineering, Friedrich-Alexander-Universität Erlangen-Nürnberg, Erlangen, Germany. [2]Erlangen Graduate School in Advanced Optical Technologies (SAOT), Erlangen, Germany. [3]Helmholtz-Institute Erlangen-Nürnberg for Renewable Energy (HI ERN), Erlangen, Germany. [4]Organic Materials & Devices, Institute of Polymer Materials, Friedrich-Alexander-Universität Erlangen-Nürnberg, Interdisciplinary Center for Nanostructured Films (IZNF), Erlangen, Germany. [5]School of Engineering, London South Bank University, London, UK. [6]Department of Electronic & Electrical Engineering, University College London, London, UK. [7]Institute of Polymer Optoelectronic Materials & Devices, Guangdong Basic Research Center of Excellence for Energy & Information Polymer Materials, State Key Laboratory of Luminescent Materials & Devices, South China University of Technology, Guangzhou, PR China. [8]Wuhan National Laboratory for Optoelectronics, Huazhong University of Science and Technology, Wuhan, PR China. ✉e-mail: c.liu@fz-juelich.de; larry.lueer@fau.de; christoph.brabec@fau.de

solar cells (OSCs), with broad spectral response, nontoxicity, effective solution processability, and superior stability, have exhibited a champion efficiency of >20% in single-junction architectures[6], positioning them as another promising candidate in the evolution of tandem photovoltaics[7].

Nowadays, integrating wide-bandgap PSCs with narrow-bandgap OSCs has emerged as a prospective route for constructing highly performing and fully solution-processed TSCs with potential for scalable manufacturing[8]. More crucially, since the solvents for dissolving perovskite and organic materials are orthogonal (dimethyl sulfoxide/dimethyl formamide and chlorobenzene/chloroform, respectively), it is less challenging to engineer fully solution production processes, compared to perovskite-perovskite tandem solar cells. That impacts as well the design criteria for a reliable interconnection layer (ICL) structure, as the requirements for solvent resistivity are more relaxed[8,9]. Thus far, perovskite-organic tandem solar cells (P-O-TSCs) have been widely studied in conventional, flexible, and semitransparent architectures[10,11]. Yet their performance still lags behind other tandem counterparts, such as silicon-perovskite and all-perovskite TSCs (34.6% and 30.1%, respectively)[5,12]. That is predominantly ascribed to substantial open-circuit voltage ($V_{OC}$) losses in both sub-cells as well as the optical/electrical losses within the ICLs[2]. Numerous strategies have been investigated to minimize the $V_{OC}$ losses in wide-bandgap PSCs, primarily through eliminating nonradiative charge recombination within the perovskite bulk or at interfaces with charge-transporting layers[3,13–19]. Likewise, the $V_{OC}$ losses in narrow-bandgap OSCs have also been continuously mitigated following intensive development efforts[6,20–22]. These achievements now shift the focus towards optimizing the ICLs, which play a pivotal role in determining the final performance of P-O-TSCs.

Ideal ICLs should demonstrate optical transparency, while simultaneously furnishing sufficient recombination sites for the extracted charges from each sub-cell[23–25]. Conventionally, metal nanoparticles (NPs) are incorporated into ICLs, forming a non-selective, metallic sink for electrons and holes to facilitate effective recombination of electrons and holes from the sub-cells. However, these noble metal NPs, such as Au and Ag, exhibit strong parasitic light absorption due to the excitation of a localized surface plasmon resonance (LSPR)[1,2]. This leads to insufficient photon utilization in the organic rear cell, thereby constraining the maximum photocurrent matching potential between the front and rear cells. In inverted (p-i-n) P-O-TSCs, it has been recently shown that metal NPs can be omitted; instead, indium tin oxide (ITO), indium oxide ($InO_x$) or indium zinc oxide (IZO) recombination layer-embedded ICLs have demonstrated excellent optical properties which led to a maximum PCE of 25.13% (23.40% certified PCE)[1,2,19]. By contrast, regular (n-i-p) P-O-TSCs are still based on metal NPs-embedded ICLs. Table S1 depicts the detailed device architectures and performance of well-performing P-O-TSCs (including p-i-n type and n-i-p type) to date. It becomes evident that all n-i-p type P-O-TSCs show significant current density losses compared to predictions based on the individual sub-cell performance, highlighting the significance of optimizing metal NP-based ICLs. Additionally, molybdenum oxide ($MoO_x$), as a preferred choice for the hole-extracting layer within the ICLs, presents challenges for long-term device stability. For instance, thermally evaporated $MoO_x$ is sensitive toward the temperature and deteriorates with annealing temperature[26], while it can also spontaneously diffuse into underlying organic layers, leading to unfavorable performance degradation over time[27–29].

In this work, we present an optimized metal-based ICL stack that combines high optical transparency with effective recombination properties, resulting in highly performing n-i-p P-O-TSCs with enhanced stability. Specifically, a solution-processed PEDOT:F layer is introduced to replace traditional $MoO_x$ in building the ICLs. Intriguingly, the different surface energies of $MoO_x$ and PEDOT:F decisively affect the diffusion behavior of gold atoms during evaporation,

resulting in the formation of Au NPs with distinct sizes and shapes. Notably, the morphological alteration of Au NPs enables to shift the plasmon resonance outside the absorption window of the low-bandgap organic absorber in the near-infrared (NIR) region. The growth of fewer Au NPs with smaller sizes and more regular spheroids is discovered on PEDOT:F surfaces. We report a significant increase in the NIR transmittance of PEDOT:F-based ICLs, as the origin for a photocurrent enhancement of >1.5 mA/cm² in precisely short-circuit current density ($J_{SC}$) matched P-O-TSCs. Concurrently, a record $V_{OC}$ of 2.32 V is delivered via combining $CsPbI_2Br$ PSC with D18-Cl:L8-BO (1.46 eV) OSC. Both improvements together allow us to realize a record champion efficiency of 25.34% for $CsPbI_2Br$/PM6:L8-BO:BTP-eC9 (1.89 eV / 1.41 eV) n-i-p type TSCs. Furthermore, the tandem devices utilizing PEDOT:F-based ICLs as well demonstrate improved stability.

## Results and discussion
### Morphology modulation of Au NPs
The detailed interconnect structures studied in this work are illustrated in Fig. 1A. Thermally evaporated 8 nm-$MoO_x$ within the ICLs is replaced by an ethanol-dispersed PEDOT:F solution that can be spin-casted (-15 nm) onto the hole transporting layer (HTL) formed by a thin D18-Cl layer (molecular structure see Fig. S1). The ICL stack is finalized by evaporating a typical 1 nm-Au layer and subsequently spin-coating layers of ZnO NPs and PFN-Br. We favor Au over Ag, as the latter is known to exhibit adverse effects on the long-term stability of the solar cell, which are related to its diffusivity and easy reaction with iodine from the perovskite bulk[30–32]. Replacing Ag with Au prevents the rapid degradation of the perovskite layer (Fig. S2). In order to understand the impact of different underlying layers on the growth of the Au nanoparticles, the surface properties of $MoO_x$ and PEDOT:F on D18-Cl were initially investigated by atomic force microscope (AFM) and contact angle measurements. As shown in Fig. S3, a higher roughness (RMS = 6.17 nm) is observed for the PEDOT:F surface compared to the $MoO_x$ surface (RMS = 1.91 nm), which derives from the solution processing of PEDOT:F via spin-coating. Furthermore, organic PEDOT:F exhibits a hydrophobic surface while the surface of $MoO_x$ is markedly hydrophilic (Fig. 1B, Fig. S4 and Table S2). Their corresponding surface free energies are calculated based on Wenzel theory, which can concurrently take the surface roughness into account (Fig. 1C; for a more detailed calculation, see Note S1, Fig. S5 and Table S3)[33,34]. The $MoO_x$ films present a high surface free energy ($\gamma_s = 28.32$ mJ/m²), comprising polar ($\gamma_s^p$) and dispersive ($\gamma_s^d$) components of 28.02 mJ/m² and 0.30 mJ/m², respectively. In contrast, PEDOT:F films exhibit distinctly low $\gamma_s$ of 6.93 mJ/m², with a significantly decreased $\gamma_s^p$ of 5.26 mJ/m² and a slightly increased $\gamma_s^d$ of 1.67 mJ/m². Typically, high surface free energy implies surface atoms are unstable and prone to interacting with other substances, while low surface free energy suggests stability, reducing the likelihood of interaction with other substances. During evaporation, Au atoms preferentially adhere to the high-surface-energy $MoO_x$ film rather than to the PEDOT:F surface, owing to a weaker interaction between atomic Au and low-surface-energy PEDOT:F film[35,36]. Additionally, the high $\gamma_s^p$ of the $MoO_x$ film may induce an electric dipole in atomic Au, leading to further electrostatic attraction[37]. This significant adsorption can reduce the diffusion rate of Au atoms on the $MoO_x$ surface, which is consistent with its lower dispersive component $\gamma_s^d$. Therefore, the stronger interaction forces between $MoO_x$ and Au atoms facilitate the attachment of Au atoms and simultaneously contribute to a retarded surface diffusion kinetics for evaporated Au atoms onto $MoO_x$ film. Conversely, their diffusion behavior on the PEDOT:F surface is diametrically opposed.

Sequentially, the ultimate morphology of Au NPs atop the $MoO_x$ and PEDOT:F was examined by scanning electron microscopy (SEM). Figure 1D, E demonstrate the tendency of Au atoms to form large and irregular Au islands on the $MoO_x$ surface, whereas smaller and more

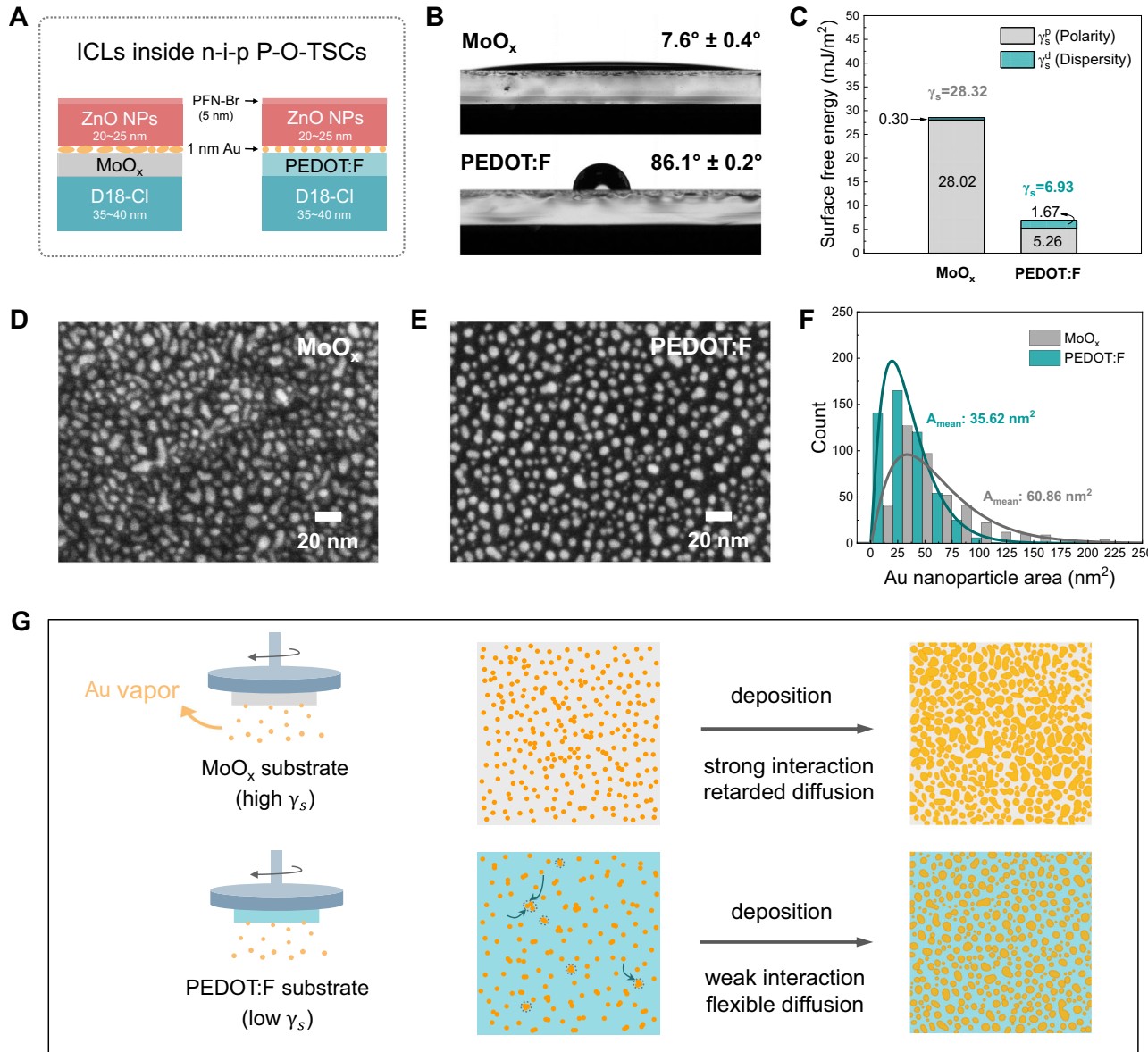

**Fig. 1 | Morphology modulation of Au NPs. A** Schematic diagram of detailed interconnect structures with thermally evaporated $MoO_x$ and solution-processed PEDOT:F, respectively. **B** Contact angle measurements with water on top of $MoO_x$ and PEDOT:F films. **C** Calculated surface free energies of $MoO_x$ and PEDOT:F films based on Wenzel theory. **D, E** SEM images of 1-nm Au NPs on the $MoO_x$ and PEDOT:F. Here, the samples for SEM measurement were based on ITO/D18-Cl/ $MoO_x$ or PEDOT:F/Au structure. The deposition rate of Au NPs was 0.02 Å/s. **F** Size distributions of 1-nm Au NPs on $MoO_x$ and PEDOT:F, which are extracted from SEM images above. **G** Schematic illustration of possible mechanism for the distinct diffusion behaviors of Au atoms onto different substrates during the evaporation process.

uniform Au particles are dispersedly formed on the organic PEDOT:F surface. These observations are quantitatively confirmed by a statistical analysis of the particle size and shape distribution from the SEM images. As displayed in Fig. 1F, Au NPs on the PEDOT:F film exhibit an average area approximately half as small ($35.62\ nm^2$) compared to those on $MoO_x$ ($60.86\ nm^2$), with a narrower size distribution. This is accompanied by more uniform spheroids, with aspect ratios closer to 1 (Fig. S6). Furthermore, the coverage of 1-nm Au NPs on PEDOT:F (33.85%) is remarkably less than on $MoO_x$ (55.51%) (Fig. S7). We denote that the smaller and more regular NP shape in combination with lower surface coverage are both expected to support efficient light propagation in the ICLs. Figure S8 provides SEM images captured under various magnifications. In order to ensure repeatability, various batches of samples underwent testing, yielding consistent results (Fig. S9).

Combining the aforementioned surface properties of $MoO_x$ and PEDOT:F layers, as well as morphology analysis of Au NPs, the possible deposition processes of Au vapor onto distinct substrate surfaces are proposed in Fig. 1G. The corresponding mechanism of diffusion behavior and morphology evolution of Au can be explained on the basis of the surface energy of the underlying substrate and its interaction with Au atoms. Regarding the underneath substrate with a high surface energy, Au atoms will show efficient surface interaction and diminished diffusion rate on it, tending to adsorb massively and spread out to form sizable and irregular Au islands, which is beneficial to reduce the system's overall free energy. Conversely, when Au vapor is deposited on an underneath surface with low surface energy (or, there is a significant disparity in surface free energy between them), weak surface interaction and high diffusion coefficient promote the movement and clustering of fewer adsorbed Au atoms. Their aggregation

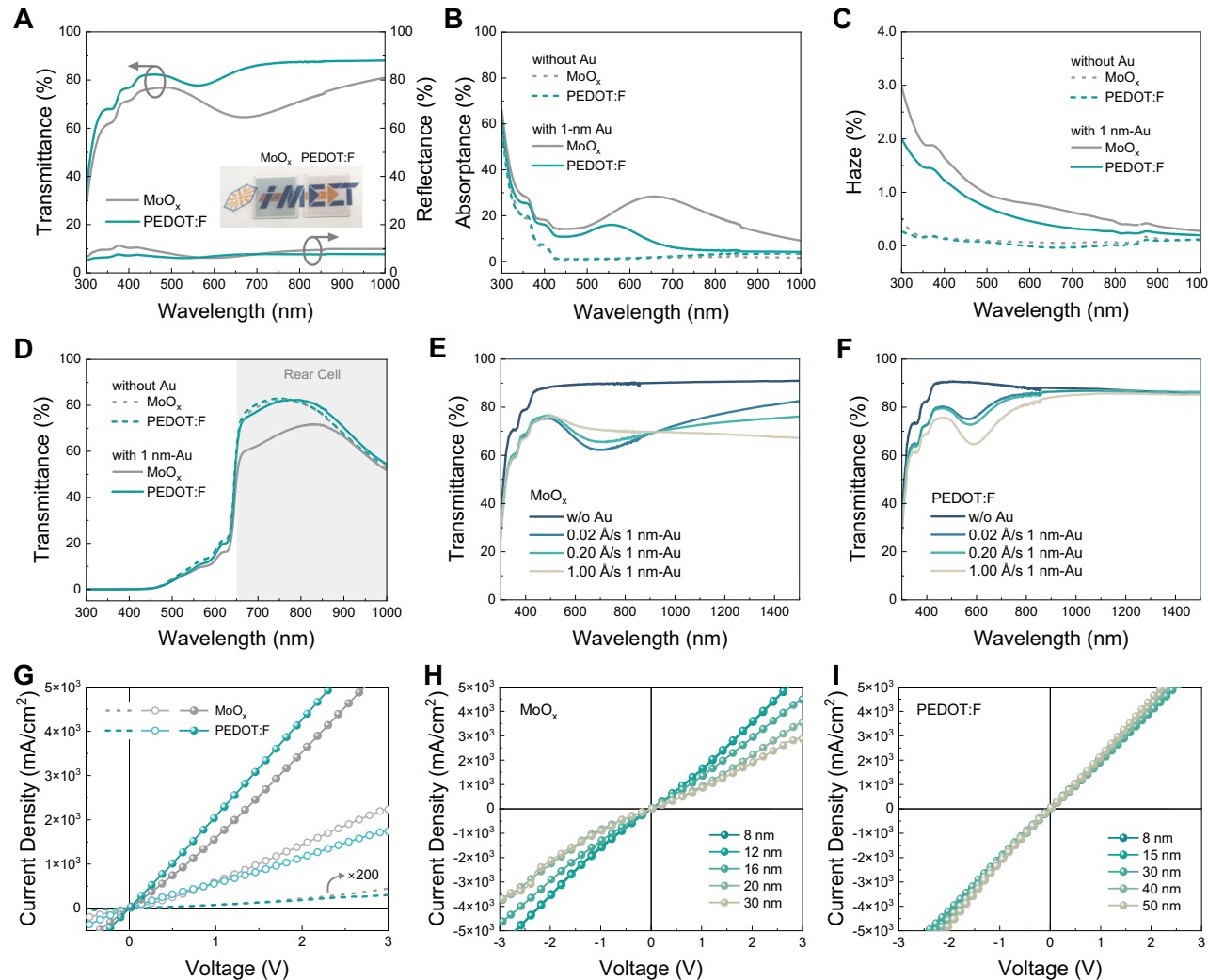

**Fig. 2 | Optical and electrical properties of Au NP-based recombination interconnect. A** Optical characteristic (total transmittance and reflectance) of glass/ MoO$_x$ or PEDOT:F/Au/ZnO/PFN-Br with an inset of their photograph (left: MoO$_x$, right: PEDOT:F). **B** Parasitic absorptance of ICLs (glass/MoO$_x$ or PEDOT:F/with or without Au/ZnO/PFN-Br). **C** Estimated haze transmittance of ICLs (glass/MoO$_x$ or PEDOT:F/with or without Au/ZnO/PFN-Br). **D** Total transmittance spectra of half-stacked-tandems integrating front perovskite cell with ICLs. **E, F** The total transmittance spectra of ICLs (glass/MoO$_x$ or PEDOT:F/with or without Au/ZnO/PFN-Br) with different deposition rates of Au recombination sites. **G** J-V characteristics of ICLs-only devices with structure of ITO/D18-Cl/MoO$_x$ or PEDOT:F/Au/ZnO/PFN-Br/ Ag (dashed line), ITO/PEDOT:PSS/D18-Cl/MoO$_x$ or PEDOT:F/Au/ZnO/PFN-Br/Ag (open-circle curve), and ITO/MoO$_x$ or PEDOT:F/Au/ZnO/PFN-Br/Ag (solid-sphere curve). The thicknesses of MoO$_x$ and PEDOT:F are 8 nm and 15 nm, respectively. First quadrant: hole injection from ITO. **H, I** J-V characteristics of ICLs-only devices (ITO/MoO$_x$ or PEDOT:F/Au/ZnO/PFN-Br/Ag) with various thicknesses of MoO$_x$ and PEDOT:F.

tends to achieve a minimal inter-surface between Au and the underneath surface for a lowest overall free energy in the whole system, yielding better uniformity of Au nanostructures.

**Optical properties of Au NP-based recombination interconnect**
The optical properties of MoO$_x$ or PEDOT:F/Au/ZnO/PFN-Br stacks on glass substrates were examined by analyzing their transmittance and reflectance spectra. As depicted in Fig. 2A, the MoO$_x$-based ICL stacks exhibit a relative low total transmittance (65–81%) for visible and NIR light. In contrast to the MoO$_x$ stack, the PEDOT:F-based ICLs display a substantial improvement in total transmittance across the entire spectrum and particularly in the NIR region (>700 nm), reaching transparency values of close to 90%. In terms of their reflectance spectra, we find negligible differences. To further clarify their distinct optical properties, UV-vis characterizations of MoO$_x$-based and PEDOT:F-based ICLs without the 1-nm Au NPs were performed. As shown in Fig. S10 and Fig. 2B, the optical response of MoO$_x$-based stacks closely resembles that of PEDOT:F-based stacks, appearing almost transparent for NIR photons. In contrast, once the nominal 1-nm thin Au NPs are

inserted, both ICLs exhibit significant but distinguished different parasitic absorption losses (Fig. 2B). We assign the optical losses to a LSPR effect of Au NPs—a unique intrinsic characteristic associated with plasmonic noble metal nanoparticles[38]. It is worth noting that the spectral shape and energetic position of the plasmonic resonance are strongly sensitive to the nanoparticle size and shape[39–41]. In Fig. 2B, the absorption peaks at 360 nm and 400 nm on the high-energy side correspond to intrinsic interband transitions in Au NPs[42–44]. In the absorption spectrum of PEDOT:F-based stacks, one typical LSPR peak centered at 555 nm is clearly visible, whereas no apparent absorption is measured in the red and NIR regions. These findings agree with the formation of more regular and symmetrical Au NPs on PEDOT:F surface, as evidenced from Fig. 1E. In contrast, the LSPR peak of MoO$_x$-based stacks is centered at 659 nm, being concomitant with a broad plasmon absorption spanning from the visible to the infrared region. This is attributed to the presence of multiple plasmon modes arising from irregular and asymmetric Au NPs with larger and more scattered aspect ratios, in line with Fig. 1D[41]. Overall, while each individual Au NP atop the MoO$_x$ film exhibits a characteristic LSPR profile, the

convolution of all LSPRs generates a broadband plasmon band from the visible to the NIR region, also providing an explanation for the darker color experimentally observed for the layer stacks and shown as inset in Fig. 2A. In summary, the PEDOT-F stacks are characterized by the blue-shifted (from 659 nm to 555 nm) and diminished (from 28% to 16%) plasmon absorption peak originating from the smaller size of fewer Au NPs, which is further supported by slightly less diffuse light-scattering (Note S2, Fig. 2C, Fig. S11)[45]. Subsequently, half-stacked-tandems with the architecture ITO/perovskite front cell/ICLs were assembled to investigate the optical transmission behavior in a more straightforward manner (Fig. 2D). As expected, the PEDOT:F-based half-stacked-tandems still present a superior transmission (59–82%) in the 650–900 nm region compared to the $MoO_x$-based half-stacked-tandems (48–72%), indicating that more NIR photons can reach the organic rear cell.

To evaluate how the transparency responds to the deposition rates of Au during thermal evaporation, three different evaporation rates of 0.02 Å/s, 0.20 Å/s, and 1.00 Å/s were applied to deposit layers with nominal 1 nm thickness as referenced by the micro-balance. As shown in Fig. 2E, with the deposition rate increasing, the transmittance of $MoO_x$-based stacks at 500-930 nm improves, while it decreases significantly at longer wavelengths. Their overall transmittance curve tends to become lower and flatter, probably owing to the more dispersed shape distribution of larger Au NPs (Fig. S12A). For PEDOT:F-based stacks, there is negligible variation in transmission for deposition rates of 0.02 Å/s and 0.20 Å/s (Fig. 2F). However, at a speedy evaporation rate of 1.00 Å/s, minimal transparency forms in the visible and NIR regions. This phenomenon can clearly be attributed to a decline in nanoparticle quality under rapid deposition conditions (Fig. S12A). Nevertheless, the deposition rates of Au barely affect the conductivity of the ICLs (Fig. S12B). Overall, the optimized PEDOT:F/Au combination consistently exhibits higher transparency. Notably, all tandems in this work were based on 0.02 Å/s-deposited Au.

The influence of Au thickness on the particle morphology and optical properties of ICLs was further investigated. As illustrated in Fig. S13, as the thickness increases, the size and irregularity of Au NPs on $MoO_x$ expand substantially, which corresponds to an intensified LSPR effect beyond 600 nm. Consequently, the overall transmittance in the NIR region becomes diminished (Fig. S14A). While the size of Au NPs on the PEDOT:F surface also increases, their shape consistently tends toward a lower aspect ratio. This induces an amplification and slight redshift of the LSPR peak at 550 nm, while preserving high transmittance in the NIR region (Fig. S14A). Remarkably, when the Au thickness is reduced to 0.5 nm, the size of the Au NPs on $MoO_x$ diminishes considerably, with more uniform shapes. This results in a substantial reduction in resonance absorption and a marked improvement in light transmittance. PEDOT:F-based ICLs similarly exhibit negligible optical losses. However, at an Au thickness of 0.5 nm, both $MoO_x$-based and PEDOT:F-based ICLs exhibit a decline in electrical performance, as shown in Fig. S14B. In this work, a nominal thickness of 1 nm is employed.

## Electrical properties of Au NP-based recombination interconnect

The electrical transport properties of the complete ICLs were evaluated from the current density-voltage (*J*-*V*) characteristics of devices with a architecture of ITO/D18-Cl/$MoO_x$ (or PEDOT:F)/Au/ZnO/PFN-Br/Ag. For simplicity, the *J*-*V* results under forward bias are emphasized (with electrons injected from the Ag electrode and holes from the ITO), as this corresponds to the direction in which carriers are transported and recombined within the ICLs of real tandem devices. As illustrated in Fig. 2G, both ICL-only devices present diode-like dark *J*-*V* characteristics in the forward direction in the first quadrant (dashed lines), suggesting a serious Schottky-type barrier contact for hole injection from the ITO electrode into the D18-Cl layer (for a more in-depth

discussion, see Figs. S15–16). After omitting the D18-Cl HTL, a linear relation in the dark *J*-*V* response of ITO/PEDOT:F/Au/ZnO/PFN-Br/Ag stacks confirms the quasi-ohmic behavior (cyan solid-sphere curve)[46]. The corresponding $MoO_x$-based stack exhibits a similar quasi-ohmic behavior, however at a slightly higher resistance[47,48]. Additionally, the $MoO_x$-based stacks exhibit a sharp deterioration in contact properties when increasing the thickness of $MoO_x$ to 30 nm, as depicted in Fig. 2H. Conversely, the conductivity of PEDOT:F-based stacks exhibits only a marginal decrease with increasing thickness of PEDOT:F, sustaining excellent ohmic behavior and conductivity even up to 50 nm thick (Fig. 2I). This underscores the robustness of this specific electrical contact for a variation of different thicknesses of PEDOT:F, which is an important asset for upscaling. Furthermore, the conductivity of the PEDOT:F stack is surprisingly insensitive to both thermal annealing and exposure to air, which are important criteria if P-O-TSC shall be processed under environmental conditions (Fig. S17). In contrast to that thermally evaporated $MoO_x$-based layers are found to be highly sensitive to elevated temperatures and critical to deteriorate during annealing steps[26].

Here, we as well inserted a PEDOT:PSS layer between ITO and D18-Cl to eliminate the unfavorable interface barrier. However, the electric conductivity of ITO/PEDOT:PSS/D18-Cl/$MoO_x$ or PEDOT:F/Au/ZnO/PFN-Br/Ag stacks is underperforming as compared to ITO/$MoO_x$ or PEDOT:F/Au/ZnO/PFN-Br/Ag (Fig. 2G), essentially due to the additional series resistance introduced by D18-Cl and PEDOT:PSS. Strikingly, the ITO/PEDOT:PSS/D18-Cl/PEDOT:F/Au/ZnO/PFN-Br/Ag stacks (cyan open-circuit curve) show reduced conductivity compared to the $MoO_x$-based stacks (gray open-circuit curve), diametrically opposed to the stacks without PEDOT:PSS and D18-Cl. That can be attributed to the imperfect interface between D18-Cl and spin-coated PEDOT:F (as discussed in detail below), which represents the most realistic nature of the complete ICLs in our tandem devices.

## Device design for perovskite-organic tandem solar cells

Figure 3A shows the complete architecture of P-O-TSCs. We initially selected D18-Cl:L8-BO with a bandgap of 1.46 eV as our organic rear absorber, as it can deliver a high $V_{OC}$ of >0.93 V[49]. The corresponding chemical structures of D18-Cl and L8-BO are displayed in Fig. S1. To identify the optimal front PSC aligned well to the chosen organic rear cell, we conducted an analysis to estimate the corresponding current and efficiency limits of P-O-TSCs when applying different PSCs. The predictive efficiency is determined by multiplying the simulated $J_{SC,TSC}$ according to the optical constants (refractive index n and attenuation coefficient k) of each layer, the $V_{OC,TSC}$, and the fill factor ($FF_{TSC}$). For a realistic prediction of electrical performance, the $V_{OC,TSC}$ is defined by summing up the record $V_{OC,PSC}$ with the best $V_{OC,osc}$ from literature, while the $FF_{TSC}$ is assumed to be a constant 80%. Optical simulation was employed to predict the optimal matched $J_{SC}$s in theory for various bandgap combinations by transfer matrix method, as shown in Fig. S18. Here, the utilized optical constants of perovskite and organic layers are presented in Fig. S19. Additionally, to approximate the achievable practical $J_{SC,TSC}$ in a real device, we take the maximum reported thicknesses of high-quality perovskite film and organic active layer from literature into consideration (for more detailed information, refer to Table S4). For instance, the theoretical maximum efficiency for $CsPbI_2Br$/D18-Cl:L8-BO P-O-TSCs can be obtained with a thickness combination of 341 nm ($CsPbI_2Br$) and 601 nm (D18-Cl:L8-BO), which is manifestly unattainable. Thus, we introduce feasible thickness combination (284 nm for $CsPbI_2Br$ and 91 nm for D18-Cl:L8-BO) to define the practical limits. As summarized in Fig. 3B and Fig. 3C, it is clear that the -1.90 eV PSC offers the best match to the D18-Cl:L8-BO rear cell across theoretical and practical assessments. In addition, for the case of the high-performance PM6:L8-BO:BTP-eC9 rear cell with a narrower bandgap of -1.41 eV[50], the same tendency persists (Figs. S20–21, Table S5). Hence, inorganic $CsPbI_2Br$ PSC with an ultra-

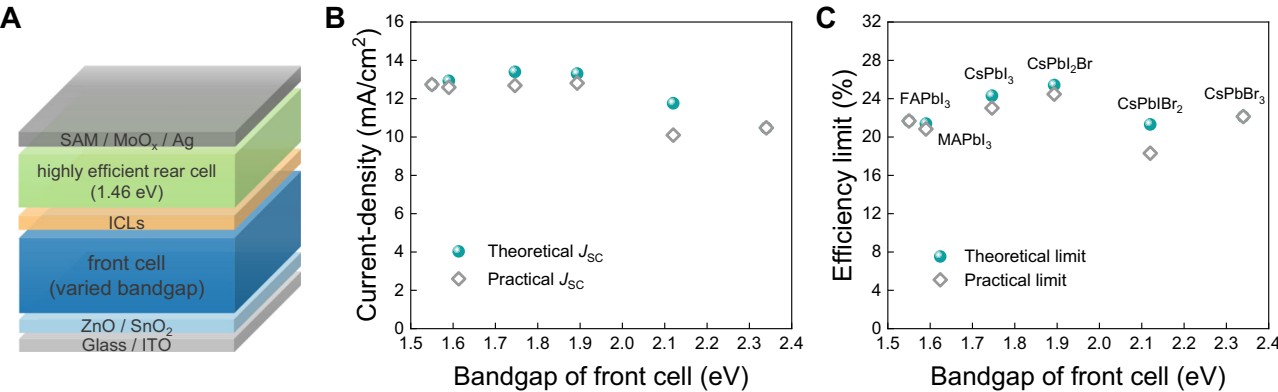

**Fig. 3 | Efficiency limits of 2-terminal perovskite-organic tandems. A** Device architecture of TSC with perovskite front sub-cell (varied bandgap) and organic rear sub-cell (1.46 eV). **B** Theoretical and practical current-density limits of TSCs based on the present high-efficient rear cell (~1.46 eV, D18-Cl:L8-BO) with different front cell bandgaps. For practical current-densities, we take the achievable thickness of high-quality perovskite film and organic active layer into account

(Table S4). **C** The corresponding theoretical and practical efficiency limits of TSCs. We assume that the $V_{OC,TSC}$ is the sum of the record $V_{OC,PSC}$, and $V_{OC,OSC}$, and the $FF_{TSC}$ is 80% (Table S4). For practical limits in a real device, the maximum reported thicknesses of high-quality perovskite film and organic active layer from literature was considered.

high $V_{OC}$ of >1.40 V is applied as a 1.89-eV-wide-bandgap front sub-cell (Fig. S22A)[13]. The corresponding single-junction devices reach a champion PCE of 17.22%, along with a $V_{OC}$ of 1.40 V, a $J_{SC}$ of 15.68 mA/cm², and a fill factor (FF) of 79%. Likewise, a PCE of 16.86% with a remarkably high $V_{OC}$ of 0.937 V and a $J_{SC}$ of 24.07 mA/cm² is achieved in inverted single-junction D18-Cl:L8-BO OSCs (Fig. S22B). Their complimentary light absorption spectra are depicted in Fig. S23.

**Perovskite-organic tandem performance**

Monolithic P-O-TSCs were fabricated with a configuration of glass/ITO/ZnO/SnO₂/CsPbI₂Br/ICLs/D18-Cl:L8-BO/Cl-2PACz/MoO$_x$/Ag to assess the performance of control and target ICLs (D18-Cl/MoO$_x$ or PEDOT:F/Au/ZnO/PFN-Br) in the complete tandem device. Figure 4A exhibits the cross-section SEM image of P-O-TSCs with the optimized thickness combination of CsPbI₂Br front cell (~285 nm) and D18-Cl:L8-BO rear cell (~93 nm), as employed in this study. The optimization process for the thickness combination of front and rear cells is shown in Figs. S24–25, coinciding with the optimum values determined by the optical simulation. Figure 4B–C present the $J$-$V$ characteristics and external quantum efficiency (EQE) curves of champion CsPbI₂Br/D18-Cl:L8-BO tandems with both ICLs, respectively. The corresponding performance parameters are summarized in Tables S6–7 and Fig. S26. Noted that an extra anti-reflection film (ARF) was applied to all tandems during $J$-$V$ and EQE measurements. As anticipated, tandem devices incorporating PEDOT:F/Au deliver a superior $J_{SC}$ of 12.66 mA/cm² compared to those utilizing MoO$_x$/Au (11.10 mA/cm²), representing a significantly raised $J_{SC}$ by >1.5 mA/cm². EQE spectra disclose that tandems employing both ICLs yield nearly identical integrated $J_{SC}$s for perovskite front cells with values of 12.68 mA/cm² vs 12.59 mA/cm². In contrast, for the integrated $J_{SC}$s of rear cells, tandem employing MoO$_x$/Au-based ICLs delivers a lower $J_{SC}$ of 10.89 mA/cm², whereas tandem utilizing PEDOT:F/Au-based ICLs realizes a superior $J_{SC}$ of 12.45 mA/cm², demonstrating a more balanced current match between two sub-cells. The enhancement of photon-harvesting occurs within the 650–900 nm range, aligning well with the observed improvement in transmittance spectra in Fig. 2D. In addition to the satisfyingly high tandem $J_{SC}$, a $V_{OC}$ value of up to 2.32 V is achieved in tandems with the optimized PEDOT:F/Au-based ICLs. This is the highest $V_{OC}$ value reported for monolithic P-O-TSCs to date (Fig. 4D) and underlines the negligible energy losses inside the ICLs. Figure 4E–F show the histograms of $V_{OC}$ and $J_{SC}$ values of 25 tandem devices from three batches. Ultimately, the best CsPbI₂Br/D18-Cl:L8-BO TSC achieves a notable PCE of 21.12% (a stable power output of 20.5%, as shown in Fig. S27),

whereas the TSC utilizing MoO$_x$/Au only shows a PCE of 19.69%. However, PEDOT:F-based tandems exhibit a lower FF of 72.53%. By comparing the real $J$-$V$ curve with the pseudo-$JV$ (p$JV$) curve derived from intensity-dependent $J$-$V$ measurements (Fig. S28), it is evident that FF losses are attributed to charge-transport losses and/or series resistance, as the p$JV$ curve only accounts for non-radiative recombination and excludes charge extraction/transport ability and series resistance[51]. Generally, the primary factors affecting charge extraction and transport ability are the carrier mobilities in the transport layers, interface barriers, and extraction rates between layers. The impact of low mobility of transporting layers could be estimated by maximizing the mobility in each layer by the drift-diffusion simulation, which is challenging to realize experimentally. As exhibited in Fig. S29, even assuming ultra-high mobility of each layer in the tandem device, there is no substantial improvement observed in the simulated FF in the limit of negligible transport losses. Additionally, in terms of the possibility of the imperfect interface between D18-Cl and PEDOT:F, the impacts of potential barriers and low transfer rates between them on P-O-TSC performance were investigated. As shown in Fig. S30, the potential barrier between D18-Cl and PEDOT:F does not significantly affect $V_{OC}$ and FF of P-O-TSCs, except in cases of severe barriers. In contrast, P-O-TSCs can exhibit a reduced FF while maintaining a high $V_{OC}$ within a lower carrier extraction rate regime from D18-Cl to PEDOT:F (Fig. S31). Yet, a low extraction rate between D18-Cl and PEDOT:F also introduces a noticeable kink at $V_{OC}$, behaving enhanced "series" resistivity, which contradicts the actual behavior of PEDOT:F-based tandems. From a circuit model perspective, both an interface barrier and a low extraction rate within the ICLs could trigger a reverse second diode, leading to non-ohmic contact behavior. Consequently, given the quasi-ohmic contact behavior of PEDOT:F-based ICLs (Fig. 2G, Fig. S16), we attribute the primary cause of the FF losses to excessive series resistance in the complete ICLs employing D18-Cl/PEDOT:F/Au/ZnO/PFN-Br (Fig. 2G, Fig. S32), particularly due to the additional resistance arising from the rough D18-Cl/PEDOT:F interface contact/morphology (Fig. S33). This conclusion aligns with the results obtained from dark $J$-$V$ curves (Fig. S34) and light-intensity dependent FF (Fig. S35) of actual tandems. To underline the relevance of our investigations of the various ICLs, we tested the impact of Au NP-free ICLs. In that case, we observe a Schottky-type barrier between the high-work-function MoO$_x$ and the low-work-function ZnO that leads to a strongly expressed S-shaped $J$-$V$ characteristic for P-O-TSCs (Fig. S36). We emphasize the necessity of integrating thin metal NPs within the ICLs for building highly efficient n-i-p P-O-TSCs.

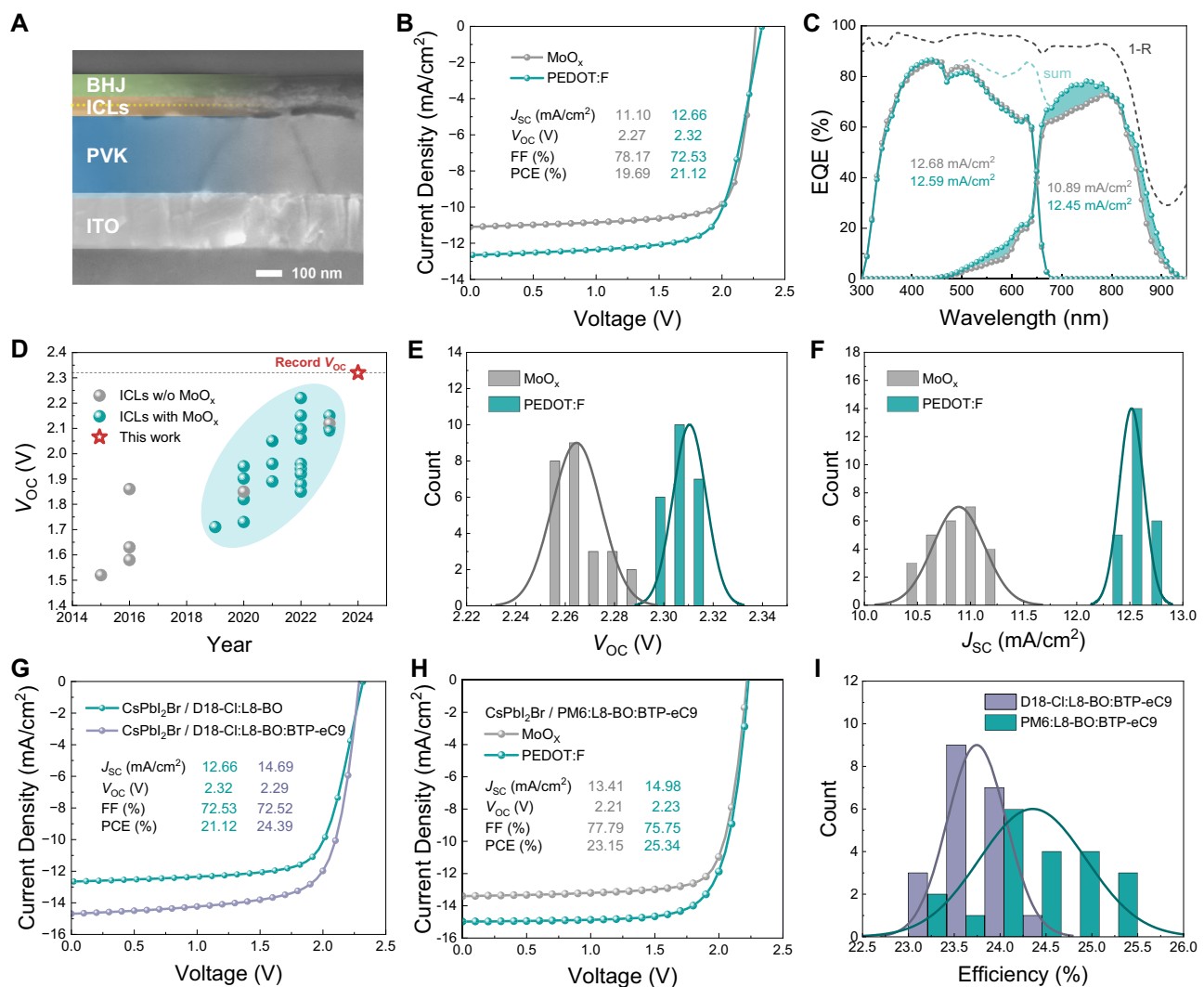

**Fig. 4 | Perovskite-organic tandem performance. A** SEM cross-section image of a TSC. **B** *J-V* characteristics of the champion CsPbI$_2$Br/D18-Cl:L8-BO P-O-TSCs with MoO$_x$-based and PEDOT:F-based ICLs. **C** Corresponding EQE spectra. Reflection (denoted as 1-R) and sum (total EQE of individual sub-cells) curves are also presented. **D** Statistical diagram of the $V_{OC}$ values obtained from the reported P-O-TSCs. **E, F** $V_{OC}$ distributions and $J_{SC}$ distributions of 25 devices over three batches. **G** *J-V* characteristics of the champion P-O-TSCs utilizing PEDOT:F, incorporating various organic rear cells (D18-Cl:L8-BO and D18-Cl:L8-BO:BTP-eC9). **H** *J-V* characteristics of the champion P-O-TSCs utilizing MoO$_x$ and PEDOT:F, incorporating highly efficient PM6:L8-BO:BTP-eC9 organic rear cells with a narrower bandgap of 1.41 eV. **I** Efficiency distributions of tandems with D18-Cl:L8-BO:BTP-eC9 and PM6:L8-BO:BTP-eC9 rear cells.

Having established the impacts of PEDOT:F/Au-based ICLs on device performance, we shifted our focus to the stability of the ICLs and these tandem cells. The PEDOT:F-based ICLs retain superior transmittance and conductivity after long-term exposure to 1 sun illumination and elevated temperature@85 °C for 750 h, as exhibited in Figs. S37–38. Interestingly, the MoO$_x$-based ICLs exhibit improved light transmittance beyond 615 nm after 750 h of photothermal exposure, likely due to external light and heat-induced changes in Au morphology, resulting in a blue shift in its LSPR peak. Their conductivity significantly decreases at 85 °C in the dark, yet remains relatively stable at 85 °C and illumination, possibly due to accelerated Au diffusion reaching ITO electrode, which could contribute to a partial "recovery" effect on conductivity. Subsequently, the performance of P-O-TSCs under dark storage and illumination conditions was monitored to assess the long-term device stability. As illustrated in Fig. 5 and Fig. S39, tandems with the PEDOT:F-based ICLs exhibit excellent stability, representing minimal degradation loss (2.11% of initial PCE) after storage under N$_2$ atmosphere for more than 4000 hours. A substantial improvement of the light operational stability in PEDOT:F-based tandems is also achieved, under continuous

metal-halide lamp (MHL) illumination with an intensity of 85 mW/cm$^2$ in a nitrogen-filled chamber at 45–50 °C (Fig. S40). Nevertheless, a significant and gradual burn-in type degradation at the first 500 h exists in both tandems. By individually exploring the device operational stability of sing-junction PSCs and OSCs (Fig. S41), it is revealed that the CsPbI$_2$Br sub-cell is the main culprit for the fast degradation, which undergoes swift degradation within initial tens of hours.

Realizing that the current tandem cells are limited in their NIR harvesting capability, we decided to add BTP-eC9 into the D18-Cl:L8-BO blend as a third component acceptor (Fig. S23). The chemical structure is displayed in Fig. S1. The single-junction ternary OSC yields an improved $J_{SC}$ of 25.18 mA/cm$^2$ and a retained high $V_{OC}$ of 0.925 V (Fig. S42). As a result, P-O-TSCs are optimized by combining a ~410 nm CsPbI$_2$Br front absorber with a ~110 nm ternary D18-Cl:L8-BO:BTP-eC9 rear absorber. That material configuration yields a champion efficiency of 24.39% (an average PCE of 23.75%), an increased $J_{SC}$ of up to 14.69 mA/cm$^2$, and a still remarkably high $V_{OC}$ of 2.29 V (Fig. 4G, Table S8). The FF remains at 72.52% without gaining. This was expected, as the previous analysis suggests the series resistance of the ICLs as the root cause for the rather low FF. The device performance of champion

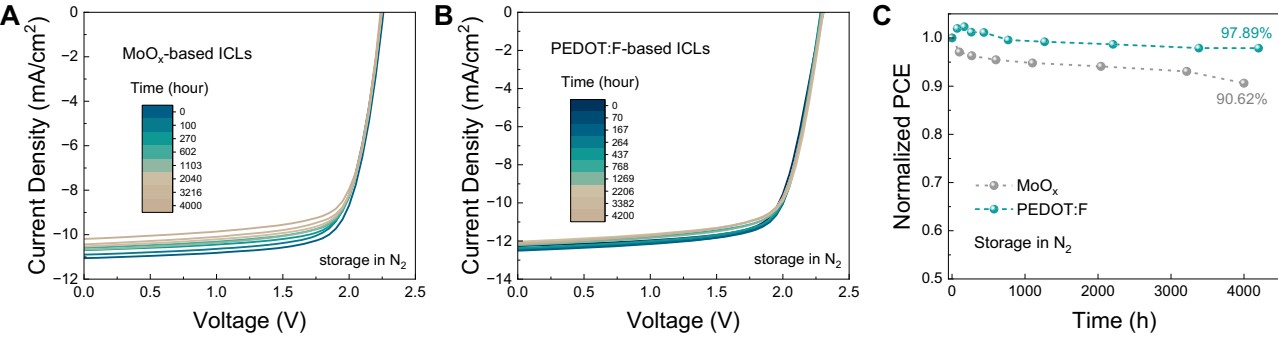

**Fig. 5 | Long-term storage stability of perovskite-organic tandems. A**, **B** *J*-*V* characteristics of CsPbI$_2$Br/D18-Cl:L8-BO P-O-TSCs employing MoO$_x$ and PEDOT:F, post-storage in N$_2$ under dark for varying durations. **C** Normalized Storage Stability of CsPbI$_2$Br/D18-Cl:L8-BO P-O-TSCs.

CsPbI$_2$Br/D18-Cl:L8-BO:BTP-eC9 TSC under reverse and forward scans is summarized in Table S9 and Fig. S43A. For the stabilized power output under maximum power point (MPP) conditions, a stable PCE of ~23.72% is achieved with a negligible performance degradation (Fig. S44). Following the same logic as the ternary OPV design, we further increased the current density by replacing D18-Cl with PM6. A small increase in the $J_{SC}$ is accompanied by a $V_{OC}$ loss of about 60 mV to a tandem $V_{OC}$ of 2.23 V. To our surprise, the PM6:L8-BO:BTP-eC9 blend effectively increases the tandem FF from about 72% to 75%. Nevertheless, PM6-based tandem devices surprisingly deliver a record efficiency of 25.34% for n-i-p type P-O-TSCs, with a $J_{SC}$ of 14.98 mA/cm$^2$, a $V_{OC}$ of 2.23 V and a FF of 75.75% (Fig. 4H, Table S10). The corresponding performance under reverse and forward scans is shown in Fig. S43B and Table S11. Meanwhile, a stable power output of ~24.12% is monitored under MPP conditions after 1000 s (Fig. S44). The histograms of PCE values of 20 tandem devices are exhibited in Fig. 4I. The spread of ±1% points is still relatively large and will require attention in further investigations. However, it is worth noting that both ternary cells, compared to the D18-Cl:L8-BO binary rear cell, demonstrate clearly enhanced EQE in the NIR range, contributing to a visible improvement in the integrated $J_{SC}$ values of champion P-O-TSCs (Fig. S45).

In summary, we develop a highly efficient ICL structure for n-i-p perovskite-organic tandem solar cells. The key to the significant performance improvement lies in the morphology control of the evaporated metal (Au) NPs layer which dominates light attenuation. We further clarify that the choice of the bottom layer is decisive to suppress extensive diffusion of Au atoms during evaporation leading to fewer, smaller, and more regular Au NPs with significantly enhanced transparency. The smaller and more spherical Au NPs on low-surface-energy PEDOT:F film exhibit inherent localized surface plasmon resonance outside the light-harvesting regime of the organic rear cell. Consequently, optimized P-O-TSCs combining CsPbI$_2$Br with various organic cells manifest a substantial current rise of >1.5 mA/cm$^2$ coming from the rear cell. Ultimately, this advancement endows efficient monolithic P-O-TSCs with a maximum PCE of 25.34% when CsPbI$_2$Br is integrated with a ternary PM6:L8-BO:BTP-eC9 composite. The significantly reduced optical losses in the ICLs bring n-i-p hybrid tandem cells to the level of p-i-n ones, providing a fresh and crucial perspective to catch up even exceed the performance of other perovskite-based tandem solar cells. Additionally, different from the p-i-n architecture, the n-i-p architecture is compatible with full-solution processing with printed back electrodes based on silver nanowires (AgNWs) or Carbon. We highlight the importance of this finding for the potential scale-up of hybrid tandems in the future.

## Methods
### Materials
Lead iodide (PbI$_2$, 99.99%) was purchased from TCI Development Co., Ltd. Cesium bromide (CsBr, ultra-dry, 99.9%), lead acetate (Pb(Ac)$_2$) and SnO$_2$ colloid (tin (IV) oxide) precursor were purchased from Alfa Aesar. ZnO NPs (zinc oxide nanoparticles dispersion, Product N-10) were received from Avantama. Polymer donor materials of D18-Cl and PM6, non-fullerene acceptors of L8-BO and BTP-eC9, and SAM molecular of 2-(3,6-Dichloro-9H-carbazol-2-yl)ethyl]phosphonic Acid (Cl-2PACz) were purchased from Solarmer Materials. Alcohol-dispersed PEDOT:F solution was synthesized based on the procedure reported by Jiang and co-workers[52]. PFN-Br was received from Sigma-Aldrich. Dimethyl sulfoxide (DMSO, extra dry, 99.70%) was purchased from Acros Organics. Isopropanol (99.50%), methanol (99.9%), Chlorobenzene (CB, 99.80%) and Chloroform (CF, 99.80%) were purchased from Sigma-Aldrich. The indium tin oxide (ITO) substrates were from Advanced Election Technology Co., Ltd. Unless stated otherwise, all materials and solvents were used as received without further purification.

### Preparation of CsPbI$_2$Br precursor solutions
0.8 M (or 1.0 M, or 1.2 M) CsPbI$_2$Br perovskite precursor solution was prepared by stoichiometrically mixing 0.8 mmol (or 1.0 mmol, or 1.2 mmol) CsBr and 0.8 mmol (or 1.0 mmol, or 1.2 mmol) PbI$_2$ in 1 mL anhydrous DMSO. Meanwhile, 13.5 mol% Pb(Ac)$_2$ additive was incorporated in order to modify the perovskite crystallinity and the energy alignment within the device[13]. This solution was stirred overnight at room temperature in the glovebox to form a clear mixture.

### Preparation of organic active solutions
Three different BHJ active layers were used to fabricate tandem solar cells. For D18-Cl:L8-BO or D18-Cl:L8-BO:BTP-eC9, the blend ratio is 1:1.2 or 1:0.65:0.65 by weight ratio, where the concentration of the D18-Cl polymer is 7 mg/ml. The donor/acceptor blend was dissolved in chloroform of 10 mg/ml 1,4-Diiodobenzene (DIB) and stirred at a 45 °C-hotplate for 4 h. For PM6:L8-BO:BTP-eC9 system, the blend ratio is 1:0.65:0.65 by weight ratio, where the concentration of the PM6 polymer is 7 mg/ml. The donor/acceptor blend was dissolved in chloroform and stirred at a 45 °C-hotplate for 4 h. 0.3% 1,8-Diiodooctane (DIO) was added into the solution 30 min earlier before use.

### CsPbI$_2$Br device fabrication
The ITO substrates were ultrasonically cleaned with acetone and isopropanol for 10 min each. Then, the substrates were dried with nitrogen gas and treated by oxygen plasma for 5 min. ZnO NPs were diluted by isopropanol (1 mL of ZnO NPs/4 mL of isopropanol) and SnO$_2$ colloid precursor was diluted by deionized (DI) water and isopropanol (1 mL of SnO$_2$ precursor/2 mL of DI water/2 mL of isopropanol). Before use, both ZnO NPs and SnO$_2$ NPs were filtered through 0.20 μm Polyamide (PA) filter. The ZnO layer and SnO$_2$ layer were sequentially spin-coated on the ITO substrate in air, with the speeds of 2500 rpm for 30 s and 3000 rpm for 30 s, respectively. Then, the substrates were

annealed at 150 °C for 30 min followed by being transferred into a nitrogen-filled glovebox. 0.8 M (or 1.0 M, or 1.2 M) CsPbI$_2$Br precursor was spun onto ITO/ZnO/SnO$_2$ substrates at 500 rpm for 10 s, 2500 rpm for 50 s, and 4000 rpm for 2 s. Subsequently, the substrates were thermally annealed by a two-step annealing process at 45 °C for 1 min and 220 °C for 1 min. During the spin-coating process, the temperature of the glovebox was kept between 24 and 27 °C. D18-Cl solution (7 mg/mL in CB with 10 mg/mL DIB additive, 45 °C) was spin-coated at 1800 rpm for 50 s. Then the n-i-p stacks were annealed at 100 °C for 5 min. Finally, the devices were finished by depositing a 10 nm-thick MoO$_x$ and a 100 nm-thick Ag electrode onto the hole-transporting layer through thermal evaporation. The effective device area was 0.038 cm$^2$.

### Organic device fabrication

The cleaned ITO substrates were oxygen plasma treated for 5 min. The undiluted and filtered ZnO NPs were spin-coated on the ITO substrate in air, with the speed of 3000 rpm for 30 s. Then, the substrates were annealed at 200 °C for 30 min followed by being transferred into a nitrogen-filled glovebox. PFN-Br (dissolved in methanol, 0.5 mg/mL) was spin-coated at 4000 rpm for 30 s onto ZnO film. After that, the organic active layers were deposited by spin-casting the organic solution on the as-prepared stacks at various speeds for 60 s. The organic films were annealed immediately under 90 °C or 100 °C for 10 min. After cooling down, the Cl-2PACz solution (0.9 mg/mL in isopropanol) was dynamically spun at 3000 rpm for 30 s. Finally, MoO$_x$ layer (10 nm) and Ag electrode (100 nm) were sequentially deposited by thermal evaporation with a deposition rate of 0.1–0.3 Å/s for MoO$_x$ and 0.1–1.0 Å/s for Ag. The effective device area was 0.038 cm$^2$.

### Fabrication of perovskite-organic tandem solar cell

For tandem, front cell containing ZnO/SnO$_2$ (electron transporting layer), perovskite and D18-Cl (hole transporting layer) was fabricated successively as the process of perovskite device fabrication. Then, MoO$_x$ (~8 nm) and Au (~1 nm) were sequentially evaporated with a deposition rate of 0.1 Å/s for MoO$_x$ and 0.02 Å/s for Au. Unless stated otherwise, all tandems were based on 0.02 Å/s-deposited Au. The alcohol-dispersed PEDOT:F solution was deposited by spin-coating at 6000 rpm for 60 s (~15 nm) to supersede the MoO$_x$ layer and annealed at 100 °C for 5 min. Subsequently, ZnO NPs (diluted by isopropanol, 1:1 v/v) and PFN-Br (dissolved in methanol, 0.5 mg/mL) were spin-coated sequentially at 4000 rpm for 30 s on the as-prepared stacks. ZnO NPs layer was annealed at 150 °C for 10 min. Refer to the organic device fabrication for the follow-up procedures.

### Film and device characterization

Current density-voltage characteristics (*J-V*) were measured with a Keithley 2400. The illumination was provided by a WAVELABS SINUS-70 3 A solar simulator with AM1.5G spectra at 100 mW cm$^{-2}$ under ambient condition. The light intensity was calibrated with a crystalline Si cell. The *J-V* characteristics were performed at a scan rate of 20 ms/step and a scan step of 50 mV. During the measurement, an anti-reflection film is applied to all tandem devices. EQE spectra of single-junction solar cells were recorded on a commercial EQE measurement system (Taiwan, Enlitech, QE-R) under ambient condition and the light intensity at each wavelength was calibrated with a standard single-crystal Si photovoltaic cell. For EQE measurement of tandem cells, the perovskite front sub-cells were measured while saturating the organic rear sub-cells with continuous light from an 808 nm laser (LDM808/3LJ), while the organic rear sub-cells were measured while saturating the perovskite front sub-cells with continuous light from a 450 nm laser (CW450-05). No bias voltage was applied for the EQE measurement of both sub-cells. During the EQE measurement, an anti-reflection film is applied to all tandem devices. The long-term operational stability of the devices was assessed at the short-circuit mode,

under continuous metal-halide lamp (MHL) illumination with an intensity of 85 mW/cm$^2$ in a nitrogen-filled chamber, with the temperature maintained at 45–50 °C. The MHL provided light in the wavelength range of 400–800 nm. Transmittance and reflectance spectra of the samples were carried out using a UV-VIS-NIR spectrometer (Lambda 950, from Perkin Elmer). Scanning electron microscopy (SEM) measurement was obtained on a JEOL JSM-7610F Schottky field-mission scanning electron microscope (Japan) with the acceleration voltage of 10 kV. Atomic force microscope (AFM) analysis was conducted in tapping mode using a Veeco Dimension 3100 microscope. A silicon tip (Radius: 7 nm) with a spring constant of 2.0 N/m and resonance frequency of 70 kHz were employed to acquire high-resolution images consisting of 2.5 × 2.5 micron. Measured AFM data were processed with Gwyddion 2.42. Contact angle measurement was conducted by an OCA 20 model (Dataphysics). Two samples were measured with test liquids and the average values were applied for surface energy calculation. All measurements were carried out under the circumstances with the temperature of 25 °C and humidity of 40%.

### Optical simulation

The optical simulation was simulated by the transfer matrix method. The optical constants of each layer applied in the simulation were obtained from the literature or simulated by NK_Finder which was reported in ref.[53]. In order to avoid the impact of LSPR effect of Au NPs, ITO/ZnO/SnO$_2$/PVK/D18-Cl/PEDOT:F/ZnO/PFN-Br/BHJ/MoO$_x$/Ag architecture was used in the optical simulation without introducing 1 nm Au. The anti-reflection film is not considered in our optical simulation model.

### Drift-diffusion simulation

The simulations were performed using SETFOS. This program numerically solves a system of three coupled equations, namely the Poisson equation, the continuity equation, and the drift-diffusion equation.

### Reporting summary

Further information on research design is available in the Nature Portfolio Reporting Summary linked to this article.

## Data availability

All data generated or analyzed during this study are included in the published article and its Supplementary Information. Data sources for the main text figures are available via Figshare at https://doi.org/10.6084/m9.figshare.27653664.v1.

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

## Acknowledgements

J.T., C.L., Z.X., S.Q., Z.P., K.Z., L.D., C.H.L., and J.Z. are grateful for the financial support from the China Scholarship Council (CSC). J.T., Z.P., L.D., and C.H.L. gratefully acknowledge funding of the Erlangen Graduate School in Advanced Optical Technologies (SAOT) by the Bavarian State Ministry for Science and Art. J.B. gratefully acknowledges funding of RTG2861-Planar carbon lattice (RTG2861) by DFG. C.L. and K.F. gratefully acknowledge funding the Helmholtz Association in the framework of the innovation platform "Solar TAP". T.J.M. acknowledges funding from a Royal Society University Research Fellowship (URF/R1/221834) and the Royal Society Research Fellows Enhanced Research Expenses (RF/ERE/221066). C.J.B. gratefully acknowledges funding by the Deutsche Forschungsgemeinschaft (DFG, German Research Foundation) under the project numbers 182849149-SFB 953, INST 90/917, and INST 90/1093-1; financial support through the "Aufbruch Bayern" initiative of the state of Bavaria (EnCN and SFF) and the Bavarian Initiative "Solar Technologies go Hybrid" (SolTech), and the grant "DFG Research Group POPULAR". N.L. acknowledges the financial support from the National Natural Science Foundation of China (52394273 and 52373179).

## Author contributions

J.T. and C.L. conceived the idea of the work and designed the project. C.L., L.L., and C.J.B. supervised the research. J.T. fabricated single-junction perovskite solar cells and tandem devices. C.L. fabricated single-junction organic solar cells and provided all organic active absorbers for tandem devices. J.T. performed the optical and electrical characterizations of ICLs and the relevant measurements of devices. K.F. provided MATLAB script adopted for optical simulation and valuable assistance in optical data analysis. A.B., Z.X., and S.Q. measured SEM images. J.B. and M.H. helped perform AFM measurements. Z.P. helped monitor the operational stability of the devices. K.Z., N.L., L.D., C.H.L., J.Z., T.D., S.S., and T.J.M. provided valuable assistance in data analysis and curation. V.M.L.C. and K.F. assisted in drift-diffusion simulations. Y.Z. provided the PEDOT:F solution. A.O. and T.H. provided the equipment support for the characterization. J.T. wrote the first draft of the manuscript. C.L., L.L., and C.J.B. revised the manuscript. All the authors revised and approved the manuscript.

## Funding

## Competing interests

The authors declare no competing interests.
