## [Peer Review File · Nature Communications]

REVIEWER COMMENTS

Reviewer #1 (Remarks to the Author):

In this manuscript, the authors utilized the low surface energy of PEDOT:F to regulate the distribution of ultrathin metal on the surface, thereby reducing the LSPR effect. This significantly enhanced the transmittance of the ICL in the near-infrared region and improved the current and efficiency of the tandem devices. The manuscript can be published if following issues are well addressed:

1. Figure 2D shows a transmittance of ~80% in the near-infrared region for the half-stacked tandem device, while the EQE (Figure S31) of the organic sub-cell exceeds 85%, which needs to be explained. Additionally, the authors should provide the EQE of single-junction D18-Cl:L8-BO:BTP-eC9 and PM6:L8-BO:BTP-eC9 OSCs, and corresponding reflection data of tandem devices.
2. The authors claim that nominal 1-nm Au shows differences in particle size and coverage on MoOx and PEDOT:F due to the surface energy. However, nominal 1-nm Au might not be the optimal value for MoOx. It is possible that the MoOx layer with the same low Au coverage can also achieve similar high transmittance and Jsc in tandem devices, then the Au coverage could be the critical point. Therefore, the authors need to illustrate the Au evaporation process from island growth to continuous film on both substrates, and provide corresponding SEM images, transmittance, and J-V characteristics of ICL layers. If authors insist the surface energy is the critical point, it is better to employ materials with different surface energy to show the relation to optical and electrical properties systematically.
3. MoOx is a typical n-type semiconductor (10.1002/adma.201201630), which achieves hole extraction by recombining electrons with photogenerated holes at the interface between MoOx and the organic layer. The typical values for VB, CB, and EF are 9.7, 6.7, and 6.9 eV, respectively. Given the MoOx layer is susceptible to contamination, if the MoOx was not transferred from the evaporation chamber to the UPS chamber in vacuum, authors should reconsider the data.
4. The authors concluded that the increase in series resistance in the ICL is the reason for the decrease in FF of the tandem devices and provided the circuit model of the tandem devices in Figure S25. In fact, when the entire ICL layer is a non-ideal ohmic contact, the ICL model needs to consider adding a reverse diode element. This is also reflected in the electrical characterization of the ICL, where the ITO/PEDOT:F/Au/ZnO/PFN-Br/Ag device shows quasi-ohmic contact, but the ITO/PEDOT:PSS/D18-Cl/PEDOT:F/Au/ZnO/PFN-Br/Ag shows a significant injection barrier, indicating that the non-ideal contact between D18-Cl and PEDOT:F hinders carrier recombination in the ICL. The increase in series resistance may only account for a small part of the FF loss. The authors need to further evaluate the contributions of this reverse diode and series resistance to the reduction in FF.
5. The manuscript only provides storage stability data. Operational stability of tandem device should also be included in the paper to demonstrate the potential of PEDOT:F based ICL.
6. Some recent important papers (10.1038/s41560-024-01451-8, 10.1038/s41560-024-01491-0, 10.1038/s41578-023-00642-1, 10.1016/j.joule.2024.06.009) should be included in this manuscript.

Reviewer #2 (Remarks to the Author):

In this manuscript, the authors report a PEDOT:F-based interconnector layer (ICL) structure designed to minimize optical parasitic absorption, thereby enhancing the performance of n-i-p perovskite-organic tandem solar cells (P-O-TSCs). The study shows that optical losses associated with Au-embedded ICLs can be significantly reduced by tailoring the shape and size distribution of Au nanoparticles through manipulation of the surface properties of the underlying layer. The PEDOT:F-based ICL exhibits low surface free energy, promoting the formation of smaller and more homogeneous spherical Au nanoparticles, which effectively mitigate optical losses while preserving efficient recombination pathways. The optimized P-O-TSCs achieved a significantly higher current density, resulting in a champion efficiency of 25.34%. This work provides valuable guidance for the design of high-performance P-O-TSCs with minimized optical losses. However, several revisions are necessary before publication.

1. The authors have demonstrated that the thermal-evaporation deposition rates of Au significantly impact the optical properties of MoO_x and PEDOT:F-ICLs. This suggests that deposition rates critically affect the morphology of Au. Therefore, it is recommended to include SEM images as a function of deposition rates to better interpret these phenomena. Additionally, a comparison of the electrical properties of ICLs as a function of deposition rates should be included.
2. Hysteresis phenomena can significantly limit the stability of perovskite solar cells (SCs). Single-junction perovskite SC subcells exhibit pronounced hysteresis characteristics. Please provide the JV curves under forward and reverse scans for the various P-O-TSCs (e.g., D18-Cl:L8-BO:BTP-eC9 and PM6:L8-BO:BTP-eC9-based). Additionally, the stable power output (SPO) under maximum power point (MPP) conditions should be provided (e.g., 10 minutes).
3. The morphology of Au is critical in determining the optical and electrical performance of ICLs. It is suggested that the authors replace the unclear SEM images of Au nanoparticles, such as those shown in Figure S9 (MoO_x) and Figures S10 (Batch 1 and 2), with clearer images.
4. The D18-Cl-based ICL demonstrates Schottky-type barrier contact for hole injection from the ITO electrode, while quasi-ohmic linear characteristics are observed when the D18-Cl hole transport layer (HTL) is omitted. Is this due to the deeper energy level of D18-Cl? Will a carrier injection barrier be formed in the single-junction perovskite SCs? The authors should provide necessary clarification and discussion on this point.
5. By tailoring the composition of the organic active layer, a clear improvement in current density was observed. The authors should provide the external quantum efficiency (EQE) response of the optimized single-junction organic solar cells (OSCs), such as D18-Cl:L8-BO:BTP-eC9 and PM6:L8-BO:BTP-eC9, along with a labeled bandgap in the manuscript. Additionally, it is suggested to include a full spectrum (e.g., 300-1000 nm) EQE response in Figure S31.

These revisions will enhance the clarity and comprehensiveness of the manuscript, providing a more robust understanding of the findings and their implications.

Reviewer #3 (Remarks to the Author):

In the manuscript titled "Overcoming Optical Losses in Thin Metal-Based Recombination Layers for Efficient n-i-p Perovskite-Organic Tandem Solar Cells," the authors address a crucial topic for tandem solar cells. The impact of thin metal-based recombination layers on optical losses is well-known, and the current consensus in the tandem community is that these losses are inevitable. Consequently, researchers have been focused on developing novel recombination layer structures to replace these thin metal nanoparticles. However, this work reports methods to minimize or eliminate these losses, offering new insights for advancing the field. While the device performances are impressive, some details in the manuscript are unclear and rarely discussed. Therefore, I recommend accepting the manuscript once the authors address the following issues to improve its quality further.

1. From the AFM measurements in Figure S4, the PEDOT:F layer exhibited a quite rough surface (RMS=6.17 nm) on top of D18-Cl. Usually, PEDOTs can form super flat films. Therefore, I am wondering whether this roughness comes from D18-Cl or insufficient interface contact. To make better understanding, the surface roughness of pure D18-Cl layer and pure PEDOT:F layer should be measured. Why does such high roughness of PEDOT:F have no obvious adverse effect on device performance?
2. To avoid misleading, the labels of Fig. S7 should be provided.
3. In the optical simulation, CsPbI₂Br with a 1.9 eV bandgap shows a perfect match with D18-Cl:L8-BO and PM6:L8-BO:BTP-eC9, resulting in the highest current density and efficiency in n-i-p structured perovskite-organic tandem cells (P-O-TSCs). However, the champion P-O-TSCs currently available are all based on ~1.80 eV perovskites. It seems to be challenging to achieve such high efficiency in this work. Specific explanations should be provided.
4. Considering that this work centers on the properties of Au-based interconnection layers, I am curious about the stability of pure ICLs. Can the authors present more measurements to evaluate the stability of pure ICLs?
5. As shown in Figures 4G and 4H, the authors further optimized the photon utilization of tandem cells by employing various organic rear cells with narrower bandgaps. However, only the EQE spectra of rear cells are provided. Since the EQE of tandem solar cells can reflect the carrier balance between the sub-cells, EQE spectra of the front perovskite cells should also be provided. In addition, discussions about the carrier balance between the sub-cells should be given.
6. The operational stability under Maximum Power Point (MPP) conditions is an essential factor in assessing the performance of solar cells. Please provide the relevant MPP stability data for the tandem devices and the single-junction sub-cells, and give detailed explanations.

REVIEWER COMMENTS

Reviewer #1 (Remarks to the Author):

In this manuscript, the authors utilized the low surface energy of PEDOT:F to regulate the distribution of ultrathin metal on the surface, thereby reducing the LSPR effect. This significantly enhanced the transmittance of the ICL in the near-infrared region and improved the current and efficiency of the tandem devices. The manuscript can be published if following issues are well addressed:

1. Figure 2D shows a transmittance of ~80% in the near-infrared region for the half-stacked tandem device, while the EQE (Figure S31) of the organic sub-cell exceeds 85%, which needs to be explained. Additionally, the authors should provide the EQE of single-junction D18-Cl:L8-BO:BTP-eC9 and PM6:L8-BO:BTP-eC9 OSCs, and corresponding reflection data of tandem devices.

2. The authors claim that nominal 1-nm Au shows differences in particle size and coverage on MoO_x and PEDOT:F due to the surface energy. However, nominal 1-nm Au might not be the optimal value for MoO_x. It is possible that the MoO_x layer with the same low Au coverage can also achieve similar high transmittance and J_{sc} in tandem devices, then the Au coverage could be the critical point. Therefore, the authors need to illustrate the Au evaporation process from island growth to continuous film on both substrates, and provide corresponding SEM images, transmittance, and J-V characteristics of ICL layers. If authors insist the surface energy is the critical point, it is better to employ materials with different surface energy to show the relation to optical and electrical properties systematically.

3. MoO_x is a typical n-type semiconductor (10.1002/adma.201201630), which achieves hole extraction by recombining electrons with photogenerated holes at the interface between MoO_x and the organic layer. The typical values for V_B, C_B, and E_F are 9.7, 6.7, and 6.9 eV, respectively. Given the MoO_x layer is susceptibility to contamination, if the MoO_x was not transferred from the evaporation chamber to the UPS chamber in vacuum, authors should reconsider the data.

4. The authors concluded that the increase in series resistance in the ICL is the reason for the decrease in FF of the tandem devices and provided the circuit model of the tandem devices in Figure S25. In fact, when the entire ICL layer is a non-ideal ohmic contact, the ICL model needs to consider adding a reverse diode element. This is also reflected in the electrical characterization of the ICL, where the ITO/PEDOT:F/Au/ZnO/PFN-Br/Ag device shows quasi-ohmic contact, but the ITO/PEDOT:PSS/D18-Cl/PEDOT:F/Au/ZnO/PFN-Br/Ag shows a significant injection barrier, indicating that the non-ideal contact between D18-Cl and PEDOT:F hinders carrier recombination in the ICL. The increase in series resistance may only account for a small part of the FF loss. The authors need to further evaluate the contributions of this reverse diode and series resistance to the reduction in FF.

5. The manuscript only provides storage stability data. Operational stability of tandem device should also be included in the paper to demonstrate the potential of PEDOT:F based ICL.

6. Some recent important papers ([10.1038/s41560-024-01451-8](https://doi.org/10.1038/s41560-024-01451-8), [10.1038/s41560-024-01491-0](https://doi.org/10.1038/s41560-024-01491-0), [10.1038/s41578-023-00642-1](https://doi.org/10.1038/s41578-023-00642-1), [10.1016/j.joule.2024.06.009](https://doi.org/10.1016/j.joule.2024.06.009)) should be included in this manuscript.

Reviewer #2 (Remarks to the Author):

In this manuscript, the authors report a PEDOT:F-based interconnector layer (ICL) structure designed to minimize optical parasitic absorption, thereby enhancing the performance of n-i-p perovskite-organic tandem solar cells (P-O-TSCs). The study shows that optical losses associated with Au-embedded ICLs can be significantly reduced by tailoring the shape and size distribution of Au nanoparticles through manipulation of the surface properties of the underlying layer. The PEDOT:F-based ICL exhibits low surface free energy, promoting the formation of smaller and more homogeneous spherical Au nanoparticles, which effectively mitigate optical losses while preserving efficient recombination pathways. The optimized P-O-TSCs achieved a significantly higher current density, resulting in a champion efficiency of 25.34%.

This work provides valuable guidance for the design of high-performance P-O-TSCs with minimized optical losses. However, several revisions are necessary before publication.

1. The authors have demonstrated that the thermal-evaporation deposition rates of Au significantly impact the optical properties of MoO_x and PEDOT:F-ICLs. This suggests that deposition rates critically affect the morphology of Au. Therefore, it is recommended to include SEM images as a function of deposition rates to better interpret these phenomena. Additionally, a comparison of the electrical properties of ICLs as a function of deposition rates should be included.

2. Hysteresis phenomena can significantly limit the stability of perovskite solar cells (SCs). Single-junction perovskite SC subcells exhibit pronounced hysteresis characteristics. Please provide the JV curves under forward and reverse scans for the various P-O-TSCs (e.g., D18-CI:L8-BO:BTP-eC9 and PM6:L8-BO:BTP-eC9-based). Additionally, the stable power output (SPO) under maximum power point (MPP) conditions should be provided (e.g., 10 minutes).

3. The morphology of Au is critical in determining the optical and electrical performance of ICLs. It is suggested that the authors replace the unclear SEM images of Au nanoparticles, such as those shown in Figure S9 (MoO_x) and Figures S10 (Batch 1 and 2), with clearer images.

4. The D18-CI-based ICL demonstrates Schottky-type barrier contact for hole injection from the ITO electrode, while quasi-ohmic linear characteristics are observed when the D18-CI hole transport layer (HTL) is omitted. Is this due to the deeper energy level of D18-CI? Will a carrier injection barrier be formed in the single-junction perovskite SCs? The authors should provide necessary clarification and discussion on this point.

5. By tailoring the composition of the organic active layer, a clear improvement in current density was observed. The authors should provide the external quantum efficiency (EQE) response of the optimized single-junction organic solar cells (OSCs), such as D18-CI:L8-BO:BTP-eC9 and PM6:L8-BO:BTP-eC9, along with a labeled

bandgap in the manuscript. Additionally, it is suggested to include a full spectrum (e.g., 300-1000 nm) EQE response in Figure S31.

These revisions will enhance the clarity and comprehensiveness of the manuscript, providing a more robust understanding of the findings and their implications.

Reviewer #3 (Remarks to the Author):

In the manuscript titled "Overcoming Optical Losses in Thin Metal-Based Recombination Layers for Efficient n-i-p Perovskite-Organic Tandem Solar Cells," the authors address a crucial topic for tandem solar cells. The impact of thin metal-based recombination layers on optical losses is well-known, and the current consensus in the tandem community is that these losses are inevitable. Consequently, researchers have been focused on developing novel recombination layer structures to replace these thin metal nanoparticles. However, this work reports methods to minimize or eliminate these losses, offering new insights for advancing the field. While the device performances are impressive, some details in the manuscript are unclear and rarely discussed. Therefore, I recommend accepting the manuscript once the authors address the following issues to improve its quality further.

1. From the AFM measurements in Figure S4, the PEDOT:F layer exhibited a quite rough surface (RMS=6.17 nm) on top of D18-Cl. Usually, PEDOTs can form super flat films. Therefore, I am wondering whether this roughness comes from D18-Cl or insufficient interface contact. To make better understanding, the surface roughness of pure D18-Cl layer and pure PEDOT:F layer should be measured. Why does such high roughness of PEDOT:F have no obvious adverse effect on device performance?
2. To avoid misleading, the labels of Fig. S7 should be provided.
3. In the optical simulation, CsPbI₂Br with a 1.9 eV bandgap shows a perfect match with D18-Cl:L8-BO and PM6:L8-BO:BTP-eC9, resulting in the highest current density and efficiency in n-i-p structured perovskite-organic tandem cells (P-O-TSCs). However, the champion P-O-TSCs currently available are all based on ~1.85 eV

perovskites. It seems to be challenging to achieve such high efficiency in this work. Specific explanations should be provided.

4. Considering that this work centers on the properties of Au-based interconnection layers, I am curious about the stability of pure ICLs. Can the authors present more measurements to evaluate the stability of pure ICLs?

5. As shown in Figures 4G and 4H, the authors further optimized the photon utilization of tandem cells by employing various organic rear cells with narrower bandgaps. However, only the EQE spectra of rear cells are provided. Since the EQE of tandem solar cells can reflect the carrier balance between the sub-cells, EQE spectra of the front perovskite cells should also be provided. In addition, discussions about the carrier balance between the sub-cells should be given.

6. The operational stability under Maximum Power Point (MPP) conditions is an essential factor in assessing the performance of solar cells. Please provide the relevant MPP stability data for the tandem devices and the single-junction sub-cells, and give detailed explanations.

Point-to-point response to the Reviewers' comments

Manuscript #: NCOMMS-24-39617-T

We sincerely thank all reviewers for the valuable suggestions, which have enabled us to substantially improve the quality of the manuscript. Detailed responses to each of the reviewers' comments are provided below.

Reviewer #1 (Remarks to the Author):

In this manuscript, the authors utilized the low surface energy of PEDOT:F to regulate the distribution of ultrathin metal on the surface, thereby reducing the LSPR effect. This significantly enhanced the transmittance of the ICL in the near-infrared region and improved the current and efficiency of the tandem devices. The manuscript can be published if following issues are well addressed:

1. Figure 2D shows a transmittance of ~80% in the near-infrared region for the half-stacked tandem device, while the EQE (Figure S31) of the organic sub-cell exceeds 85%, which needs to be explained. Additionally, the authors should provide the EQE of single-junction D18-Cl:L8-BO:BTP-eC9 and PM6:L8-BO:BTP-eC9 OSCs, and corresponding reflection data of tandem devices.

Response: We thank the reviewer for highlighting these important points. Firstly, the discrepancy between the transmittance in Figure 2D and the EQE value of the organic rear cells is primarily caused due to the presence or absence of an anti-reflection film during measurements. All optical spectra were obtained without the use of anti-reflection films. However, as noted in the “**Film and device characterization**” section of the **Methods** (Page 24, lines 529 and 537), an anti-reflection film was applied to all tandem devices during the *JV* and EQE measurements. The use of the anti-reflective film during the EQE measurement increased the light absorption of rear organic solar cells by reducing the reflectance of the ITO glass substrates. Consequently, the number of free charges collected by the device to the external circuit increased, resulting in an enhancement of the EQE spectra.

To prevent any potential confusion, we have added the following explanation in “Perovskite-organic tandem performance” section of the main text:

Page 15, line 337 of the text.

“Noted that, an extra anti-reflection film was applied to all tandems during *J-V* measurements.”

Secondly, we have supplemented the respective EQE curves for the single-junction OSCs based on D18-Cl:L8-BO:BTP-eC9 and PM6:L8-BO:BTP-eC9 (without the anti-reflection film) and the respective reflection data for the tandem devices (with the anti-reflection film) in the revised SI and main text, as shown below:

SI_Page 32, Figure S42.

“

Figure S42. Device performance and the corresponding EQE spectra of single-junction (A) D18-Cl:L8-BO:BTP-eC9 and (B) PM6:L8-BO:BTP-eC9 OSCs.”

SI_Page 34, Figure S45.

“

Figure S45. Corresponding EQE spectra of the champion P-O-TSCs utilizing PEDOT:F, incorporating D18-Cl:L8-BO:BTP-eC9 and PM6:L8-BO:BTP-eC9 organic rear cells. Reflection (denoted as 1-R) and sum (total EQE of individual sub-cells) curves are also presented.”

Page 37, Figure 4C in the main text.

“

Figure 4. (C) Corresponding EQE spectra. Reflection (denoted as 1-R) and sum (total EQE of individual sub-cells) curves are also presented.”

2. The authors claim that nominal 1-nm Au shows differences in particle size and coverage on MoO_x and PEDOT:F due to the surface energy. However, nominal 1-nm Au might not be the optimal value for MoO_x. It is possible that the MoO_x layer with the same low Au coverage can also achieve similar high transmittance and J_{sc} in tandem devices, then the Au coverage could be the critical point. Therefore, the authors need to illustrate the Au evaporation process from island growth to continuous film on both substrates, and provide corresponding SEM images, transmittance, and J-V characteristics of ICL layers. If authors insist the surface energy is the critical point, it is better to employ materials with different surface energy to show the relation to optical and electrical properties systematically.

Response: We sincerely thank the reviewer for your valuable comment. Yes, our research results demonstrate that the surface properties of the underlying layer can decisively influence the morphology of Au NPs, including their size, shape, and coverage. We fully agree that the Au coverage is a critical factor affecting the transmittance of ICLs, with higher coverage leading to lower transmittance and lower coverage resulting in higher transmittance.

Accordingly, we intended to illustrate the Au evaporation process from island growth to continuous film by applying various nominal thicknesses of 0.5 nm, 1.0 nm, 1.5 nm, 2.0 nm, and 2.5 nm, and investigated the effect of Au thickness on partial morphology, as well as the optical and electrical properties of ICLs. Relevant analysis of these resultant data is shown below.

1. From the aspects of morphology and optical properties: With the thickness increases from 0.5 nm to 2.5 nm, the size and irregularity of Au NPs on MoO_x increase substantially (Figure S13), corresponding to a significantly intensified localized surface plasmon resonance (LSPR) effect beyond 600 nm. Consequently, the overall transmittance spectra in the NIR region are steeply reduced and maintain flat (Figure S14A). While the size of Au NPs on the PEDOT:F surface also increases, their shape consistently tends toward a lower aspect ratio. This induces an amplification and slight redshift of the LSPR peak at 550 nm, while maintaining high transmittance in the NIR region (Figure S14A). It is worth pointing out that, lowering the Au coverage by reducing the thickness of the Au film to 0.5 nm indeed improves the transmittances of both ICLs. Therefore, from an optical perspective, a nominal Au thickness of 1 nm is not optimal for the MoO_x-based ICLs, nor for the PEDOT:F-based ICLs. PEDOT:F-based ICLs exhibit no optical loss beyond 650 nm and maintain significantly higher overall transmittance than MoO_x-based ICLs at the same 0.5-nm Au. This highlights the crucial role of the underlying layer in the formation of Au NPs, even though the morphological differences in Au are difficult to observe under ultra-low Au thickness condition.

2. From the aspect of electrical properties: Figure S14B exhibits that thicker Au film with higher coverage within the ICL is beneficial for the electrical contact. However, it is observed that both MoO_x-based and PEDOT:F-based ICLs embedded with an ultrathin Au thickness of 0.5 nm show reduced electrical performance, as there are insufficient recombination sites to effectively promote the recombination of electrons and holes.

From the above discussion, it is clearly evidenced that the optical and electrical properties of Au-based ICLs needs to be well balanced. By combining the overall results as shown in Figures S13-S14, PEDOT:F-based ICLs exhibit more favorable optical and electrical performance across a broader range of processing conditions than MoO_x-based ICLs. Most importantly, these new results indicate that the optimal thickness of the evaporated Au film on the PEDOT:F layer lies between 0.5 nm and 1.0 nm, providing valuable guidance for future studies.

Furthermore, the reviewer's suggestion to explore the relationship between surface energy and optical/electrical properties using materials with varying surface energies is insightful. However, we believe that randomly selecting materials would introduce more unpredictable influencing factors. On the other hand, we are currently unable to systematically investigate this using a range of PEDOT:F materials with different surface energies. This is because synthesizing PEDOT:F materials with tailored surface energies presents a considerable challenge within a limited timeframe. Nevertheless, we will keep this suggestion in mind as an important direction for our follow-up research. However, for a simple comparison, we still chose a similar commercial PEDOT:PSS (AI 4083) material, with the surface energy value falling between those of MoO_x and PEDOT:F. Accordingly, at a nominal thickness of 1 nm, Au forms relatively regular small particles, leading to improved optical properties compared to MoO_x-based ICLs, as displayed in *Figure_PEDOT:PSS* below.

SI_Page 14 and Page 15, Figures S13 and S14.

“

Figure S13. SEM images at 250,000x and 500,000x magnifications of Au NPs deposited on (A) MoO_x and (B) PEDOT:F with varying thicknesses (0.5 nm, 1.0 nm, 1.5 nm, 2.0 nm, 2.5 nm). Here, all samples for SEM measurement were based on ITO/MoO_x or PEDOT:F/Au structure.

Figure S14. (A) The total transmittance spectra of ICLs (glass/MoO_x or PEDOT:F/Au/ZnO/PFN-Br) with varying Au thicknesses. (B) The corresponding *J-V*

curves of ICLs (ITO/MoO_x or PEDOT:F/Au/ZnO/PFN-Br/Ag) with varying Au thicknesses.”

Page 11, line 247 of the main text.

“The influence of Au thickness on the particle morphology and optical properties of ICLs was further investigated. As illustrated in Figure S13, with increasing the thickness, the size and irregularity of Au NPs on MoO_x expand substantially, which corresponds to an intensified LSPR effect beyond 600 nm. Consequently, the overall transmittance in the NIR region becomes diminished (Figure S14A). While the size of Au NPs on the PEDOT:F surface also increases, their shape consistently tends toward a lower aspect ratio. This induces an amplification and slight redshift of the LSPR peak at 550 nm, while preserving high transmittance in the NIR region (Figure S14A). Remarkably, when the Au thickness is reduced to 0.5 nm, the size of the Au NPs on MoO_x diminishes considerably, with more uniform shapes. This results in a substantial reduction in resonance absorption and a marked improvement in light transmittance. PEDOT:F-based ICLs similarly exhibit negligible optical losses. However, at an Au thickness of 0.5 nm, both MoO_x-based and PEDOT:F-based ICLs exhibit a decline in electrical performance, as shown in Figure S14B. In this work, a nominal thickness of 1 nm is employed.”

Figure_PEDOT:PSS

3. MoO_x is a typical n-type semiconductor (10.1002/adma.201201630), which achieves hole extraction by recombining electrons with photogenerated holes at the

interface between MoO_x and the organic layer. The typical values for VB, CB, and E_F are 9.7, 6.7, and 6.9 eV, respectively. Given the MoO_x layer is susceptible to contamination, if the MoO_x was not transferred from the evaporation chamber to the UPS chamber in vacuum, authors should reconsider the data.

Response: Many thanks for the reviewer's insightful comment. We have reviewed the relevant literature and agree that the work function of MoO_x is indeed highly sensitive to fabrication and testing environments. As noted, the E_F of a freshly evaporated MoO_x film obtained under vacuum condition is 6.9 eV. However, ambient exposure can lower the E_F of MoO_x (e.g. 5.7 eV) due to changes in chemical composition at the surface, such as chemisorbed oxygen and impurity adsorption (<https://doi.org/10.1063/1.4748978>). In terms of our measurement, MoO_x film was deposited via thermally-evaporation under vacuum, and the PEDOT:F film was spin-coated. However, they were packaged in a nitrogen glovebox and then express-delivered from Germany to UK for UPS testing at another laboratory. As there was a significant time interval between preparation and testing, it is possible that the samples were contaminated during packaging, transporting, and measuring, which caused the discrepancies between our results and those obtained under vacuum conditions. Unfortunately, we are currently unable to perform the UPS measurements from the evaporation chamber or a clean glovebox to the UPS chamber in vacuum directly. To avoid any potential misinterpretation, we have removed the following expression and data from the main text and SI:

Page 6, line 136 of the main text.

~~“(higher work function, Figure S1)”~~

SI_Page 7, Figure S1.

~~“~~

Figure S1. Photoemission cutoff spectra (Left) and valence band spectra (Right) of (A) MoO_x and (B) PEDOT:F films, measured by ultraviolet photoemission spectroscopy (UPS).”

4. The authors concluded that the increase in series resistance in the ICL is the reason for the decrease in FF of the tandem devices and provided the circuit model of the tandem devices in Figure S25. In fact, when the entire ICL layer is a non-ideal ohmic contact, the ICL model needs to consider adding a reverse diode element. This is also reflected in the electrical characterization of the ICL, where the ITO/PEDOT:F/Au/ZnO/PFN-Br/Ag device shows quasi-ohmic contact, but the ITO/PEDOT:PSS/D18-Cl/PEDOT:F/Au/ZnO/PFN-Br/Ag shows a significant injection barrier, indicating that the non-ideal contact between D18-Cl and PEDOT:F hinders carrier recombination in the ICL. The increase in series resistance may only account for a small part of the FF loss. The authors need to further evaluate the contributions of this reverse diode and series resistance to the reduction in FF.

Response: We thank the reviewer for this valuable suggestion on the FF loss analysis of tandem devices. We agree that the potential influence of a reverse diode introduced by the ICLs should be considered in the FF loss discussion. In the original manuscript, by comparing the real *J-V* curve with the pseudo-*JV* (*pJV*) curve derived from intensity-dependent *J-V* measurements (Figure S28), it has been evident that FF losses in our tandem devices are attributed to charge-transport losses and/or series resistance, as the *pJV* curve only accounts for non-radiative recombination and excludes charge extraction/transport ability and series resistance.

In general, the primary factors affecting charge extraction and transport ability include the carrier mobilities of the transport layers, interface barriers, and extraction rates between layers. Therefore, each of these factors is examined individually.

Firstly, the impact of low mobility of transporting layers could be estimated by maximizing the mobility in each layer by the drift-diffusion simulation, which is difficult to realize experimentally. It can be seen from the simulated results (Figure S29) that even assuming ultra-high mobility of each layer in the tandem device, there is no substantial improvement observed in the simulated FF in the limit of negligible transport losses, which rules out the mobility as a factor in FF losses.

We then focus on other potential factors: interface barriers and low extraction rates between layers. From a circuit model perspective, both an interface barrier and a low extraction rate within the ICLs could trigger a reverse second diode, leading to non-ohmic contact behavior. As demonstrated in Figure 2G, the ICL stack with ITO/PEDOT:F/Au/ZnO/PFN-Br/Ag structure shows superior quasi-ohmic contact, while the ITO/PEDOT:PSS/D18-Cl/PEDOT:F/Au/ZnO/PFN-Br/Ag stack shows inferior electrical properties. However, it still demonstrates quasi-ohmic contact, despite not being perfectly linear. Here, for simplicity, we only focus on the J - V results when the forward bias is applied (first quadrant) as this corresponds to the direction in which carriers are transported and recombined in the ICLs of real tandem devices (electrons are injected from the Ag electrode, and holes are injected from the ITO). This approach reflects actual device behavior. Therefore, we do not think there is a significant injection barrier or low extraction rate between D18-Cl and PEDOT:F.

To confirm this, we simulate the impacts of potential barriers and extraction rates between D18-Cl and PEDOT:F on P-O-TSC performance, respectively. As shown in Figure S30, varying the potential barrier between D18-Cl and PEDOT:F does not significantly affect V_{oc} and FF of P-O-TSCs, except in cases of severe barriers (> 0.35 eV), which is obviously not consistent with the real experimental tandem cases. Interestingly, Figure S31 reveals that within a lower carrier extraction rate regime from D18-Cl to PEDOT:F, P-O-TSCs can maintain high V_{oc} but exhibit reduced FF, consistent with the device performance we observed. Yet, a low extraction rate between D18-Cl and PEDOT:F also introduces a noticeable kink at V_{oc} , behaving enhanced 'series' resistivity, which contradicts the actual behavior of PEDOT:F-based tandems. It is noted out that the extraction rate value between the two layers cannot

be directly adjusted in Setfos. Therefore, we simulated a reduced extraction rate from D18-Cl to PEDOT:F by lowering the mobility of PEDOT:F.

Consequently, given the quasi-ohmic contact behavior of PEDOT:F-based ICLs (Figure 2G and Figure S16), we still attribute the primary cause of FF losses to excessive classical series resistance in the complete ICLs employing D18-Cl/PEDOT:F/Au/ZnO/PFN-Br (Figure 2G, Figure S32), particularly due to the additional resistance introduced by the inferior D18-Cl/PEDOT:F interface (Figure S33). We have included this detailed discussion and supplemental data in the revised manuscript and SI, as shown below:

Page 16, line 354 of the text.

“By comparing the real J - V curve with the pseudo- JV (pJV) curve derived from intensity-dependent J - V measurements (Figure S28), it is evident that FF losses are attributed to charge-transport losses and/or series resistance, as the pJV curve only accounts for non-radiative recombination and excludes charge extraction/transport ability and series resistance.⁵¹ Generally, the primary factors affecting charge extraction and transport ability are the carrier mobilities in the transport layers, interface barriers, and extraction rates between layers. The impact of low mobility of transporting layers could be estimated by maximizing the mobility in each layer by the drift-diffusion simulation, which is challenging to realize experimentally. As exhibited in Figure S29, even assuming ultra-high mobility of each layer in the tandem device, there is no substantial improvement observed in the simulated FF in the limit of negligible transport losses. Additionally, in terms of the possibility of the imperfect interface between D18-Cl and PEDOT:F, the impacts of potential barriers and low transfer rates between them on P-O-TSC performance were investigated. As shown in Figure S30, the potential barrier between D18-Cl and PEDOT:F does not significantly affect V_{oc} and FF of P-O-TSCs, except in cases of severe barriers. In contrast, P-O-TSCs can exhibit a reduced FF while maintaining a high V_{oc} within a lower carrier extraction rate regime from D18-Cl to PEDOT:F (Figure S31). Yet, a low extraction rate between D18-Cl and PEDOT:F also introduces a noticeable kink at V_{oc} , behaving enhanced ‘series’ resistivity, which contradicts the actual behavior of PEDOT:F-based tandems. From a circuit model perspective, both an interface barrier and a low extraction rate within the ICLs could trigger a reverse second diode, leading to non-ohmic contact behavior. Consequently, given the quasi-ohmic contact behavior

of PEDOT:F-based ICLs (Figure 2G and Figure S16), we attribute the primary cause of the FF losses to excessive series resistance in the complete ICLs employing D18-Cl/PEDOT:F/Au/ZnO/PFN-Br (Figure 2G, Figure S32), particularly due to the additional resistance arising from the rough D18-Cl/PEDOT:F interface contact/morphology (Figure S33). This conclusion aligns with the results obtained from dark J - V curves (Figure S34) and light-intensity dependent FF (Figure S35) of actual tandems.”

Page 12, line 264 of the text.

“For simplicity, the J - V results under forward bias are emphasized (with electrons injected from the Ag electrode and holes from the ITO), as this corresponds to the direction in which carriers are transported and recombined within the ICLs of real tandem devices.”

SI_Pages 24-27, Figures S28-S33.

“

Figure S28. The pseudo- JV curves of CsPbI₂Br/D18-Cl:L8-BO TSCs employing MoO_x and PEDOT:F, which are created from intensity-dependent J - V measurements.

Figure S29. Simulated *JV*-curves of CsPbI₂Br/D18-Cl:L8-BO TSCs with PEDOT:F-based ICLs, using Setfos software. Based on the standard simulation (pink) with a FF of 73.85%, we increased the carrier mobilities of the individual layers to lower the charge transport losses. The graph shows that a 10/100/1000-fold increase of the mobility has no effect on the obvious FF.

Figure S30. (A) The impact of energy difference (ΔE) between D18-Cl and PEDOT:F on the *JV*-curves of CsPbI₂Br/D18-Cl:L8-BO TSCs with PEDOT:F-based ICLs, simulated using Setfos software. (B) Simulated V_{oc} and FF as a function of ΔE .

Figure S31. (A) The impact of the extraction rate from D18-Cl to PEDOT:F on the *JV*-curves of CsPbI₂Br/D18-Cl:L8-BO TSCs with PEDOT:F-based ICLs, simulated using Setfos software. Since the extraction rate value between the two layers cannot be directly adjusted in Setfos, we simulated a reduced extraction rate from D18-Cl to PEDOT:F by lowering the mobility of PEDOT:F. (B) Simulated V_{oc} and FF as a function of the extraction rate from D18-Cl to PEDOT:F.

Figure S32. (A) The impact of varied series resistance (R_s) on *JV*-curves of CsPbI₂Br/D18-Cl:L8-BO TSCs with PEDOT:F-based ICLs, obtained from the Setfos simulation. (B) The equivalent circuits of single-junction PSC, single-junction OSC and P-O-TSC. (C) The simulated *JV*-curves of CsPbI₂Br/D18-Cl:L8-BO TSCs with PEDOT:F-based ICLs, based on $R_{s,PSC}$, $R_{s,OSC}$ and $R_{s,PSC} + R_{s,OSC}$. Here, $R_{s,PSC}$ and $R_{s,OSC}$ are the series resistances of single-junction PSC and single-junction OSC, respectively. It is evident that the series resistance introduced by front and rear cells has slightly adverse effect on tandem's FF (pink shade), suggesting that the FF value is primarily subject to the series resistance introduced by PEDOT:F-based ICLs (blue shade).

Figure S33. AFM images of silicon wafer (Si), Si/D18-Cl, Si/MoO_x, Si/PEDOT:F, Si/D18-Cl/MoO_x and Si/D18-Cl/PEDOT:F. The pure D18-Cl layer demonstrates a relatively low roughness of 2.36 nm, while the MoO_x layer exhibits negligible roughness. In contrast, the PEDOT:F layer shows substantial roughness, measuring 5.42 nm. These findings confirm that the pronounced roughness of PEDOT:F arises from its intrinsic material properties. We hypothesize that the highly rough PEDOT:F layer forms incomplete layer-to-layer contact with D18-Cl, potentially creating microscopic voids or gaps at the interface, which adversely impacts the performance of PEDOT:F-based P-O-TSCs.”

5. The manuscript only provides storage stability data. Operational stability of tandem device should also be included in the paper to demonstrate the potential of PEDOT:F based ICL.

Response: We appreciate the reviewer’s suggestion. Operational stability is a crucial indicator for evaluating the performance of solar cells. In light of your suggestion, we have supplemented the long-term operational stability data of tandems with different ICLs at the short-circuit mode, under continuous metal-halide lamp (MHL) illumination with an intensity of 85 mW/cm² in N₂ at 45-50°C. As anticipated, the tandem devices utilizing PEDOT:F-based ICLs exhibited significantly enhanced stability (60% after 750 hours) compared to those employing MoO_x-based ICLs (33% after 750 hours). However, we noticed a significant and gradual burn-in type degradation for both

tandems at the first 500 hours, which is primarily attributed to the degradation of CsPbI₂Br perovskite front cells. We now include these stability data and a detailed discussion in the revised manuscript and SI, as shown below:

SI_Page 31, Figure S40 and Figure S41.

“

Figure S40. Normalized long-term operational stability of CsPbI₂Br-D18-Cl:L8-BO P-O-TSCs employing MoO_x and PEDOT:F at the short-circuit mode, under continuous metal-halide lamp (MHL) illumination with an intensity of 85 mW/cm² in N₂ at 45-50°C.

Figure S41. Normalized long-term operational stability of single-junction D18-CI:L8-BO OSC and CsPbI₂Br PSC at the short-circuit mode, under continuous metal-halide lamp (MHL) illumination with an intensity of 85 mW/cm² in N₂ at 45-50°C.”

Page 18, line 399 of the main text.

“A substantial improvement of the light operational stability in PEDOT:F-based tandems is also achieved, under continuous metal-halide lamp (MHL) illumination with an intensity of 85 mW/cm² in a nitrogen-filled chamber at 45-50°C (Figure S40). Nevertheless, a significant and gradual burn-in type degradation at the first 500 hours exists in both tandems. By individually exploring the device operational stability of single-junction PSCs and OSCs (Figure S41), it is revealed that the CsPbI₂Br sub-cell is the main culprit for the fast degradation, which undergoes swift degradation within initial tens of hours.”

Page 24, line 538 of the text.

“The long-term operational stability of the devices was assessed at the short-circuit mode, under continuous metal-halide lamp (MHL) illumination with an intensity of 85 mW/cm² in a nitrogen-filled chamber, with the temperature maintained at 45-50°C. The MHL provided light in the wavelength range of 400 to 800 nm.”

6. Some recent important papers (10.1038/s41560-024-01451-8, 10.1038/s41560-024-01491-0, 10.1038/s41578-023-00642-1, 10.1016/j.joule.2024.06.009) should be included in this manuscript.

Response: We thank the reviewer for highlighting these significant pieces of literature. All the suggested works focus on perovskite/organic tandem solar cells and have successfully achieved remarkable efficiencies of over 25% for p-i-n P-O-TSCs, representing the pinnacle of this field at the current stage. We have added these citations to our revised manuscript and included these high-performing P-O-TSCs in the summarized table (Table S1) in the revised SI, as shown below:

Pages 27-28, line 606 of the main text.

“17 Wu, S. *et al.* Redox mediator-stabilized wide-bandgap perovskites for monolithic perovskite-organic tandem solar cells. *Nat. Energy* **9**, 411-421 (2024).

18 Zhang, Z. *et al.* Suppression of phase segregation in wide-bandgap perovskites with thiocyanate ions for perovskite/organic tandems with 25.06% efficiency. *Nat. Energy* **9**, 592-601 (2024).

19 Guo, X. *et al.* Stabilizing efficient wide-bandgap perovskite in perovskite-organic tandem solar cells. *Joule*, doi:10.1016/j.joule.2024.06.009 (2024).

25 Brinkmann, K. O. *et al.* Perovskite-organic tandem solar cells. *Nat. Rev. Mater.* **9**, 202-217 (2024).”

SI_Pages 4-5, Table S1.

“Table S1. Device architectures and performance of well-performing P-O-TSCs.

	Device Structure	E_g (eV)	V_{oc} (V)	J_{sc} (mA/cm ²)	FF (%)	PCE (%)	Ref.
p-i-n PSC	ITO/DC-PA/Cs _{0.2} FA _{0.8} Pb(I _{0.6} Br _{0.4}) ₃ /PI/C ₆₀ /BCP/Ag	1.81	1.351	17.52	82.74	19.58	12
p-i-n OSC	ITO/MoO _x /PM6:Y6:PC71BM/PNDIT-F3N/Ag	1.33	0.840	26.58	74.43	16.62	(2024)
p-i-n TSC	ITO/DC-PA/PVK/PI/C ₆₀ /BCP/Au/MoO _x /BHJ/PNDIT-F3N/Ag		2.151	14.36	81.65	25.22	
p-i-n PSC	ITO/NiO _x /2PACz/FA _{0.8} Cs _{0.2} Pb _{1.6} Br _{1.4} -Pb(SCN) ₂ /PEAI/C ₆₀ /BCP/Ag	1.84	1.32	17.06	84.21	18.96	13
p-i-n OSC	ITO/MoO _x /2PACz/D18-Cl:N3:PC ₆₁ BM/C ₆₀ /BCP/Ag	1.38	0.856	28.08	78.41	18.85	(2024)
p-i-n TSC	ITO/NiO _x /2PACz/PVK/C ₆₀ /BCP/Ag/MoO _x /2PACz/BHJ/C ₆₀ /BCP/Ag		2.12	14.68	82.97	25.82	
p-i-n PSC	ITO/NiO _x /Me-4PACz/Cs _{0.25} FA _{0.75} Pb(I _{0.5} Br _{0.5}) ₃ /PCBM/BCP/Ag	1.86	1.366	16.10	84.20	18.52	14
p-i-n OSC	ITO/MoO _x /PM6:BTP-eC9/PDINN/Ag	1.39	0.852	26.15	74.89	16.68	(2024)
p-i-n TSC	ITO/NiO _x /Me-4PACz/PVK/PCBM/AZO/ITO/MoO _x /BHJ/PDINN/Ag		2.144	14.65	80.02	25.13	

SI_Page 35, Supplemental References.

“12 Wu, S. *et al.* Redox mediator-stabilized wide-bandgap perovskites for monolithic perovskite-organic tandem solar cells. *Nat. Energy* **9**, 411-421 (2024).

13 Zhang, Z. *et al.* Suppression of phase segregation in wide-bandgap perovskites with thiocyanate ions for perovskite/organic tandems with 25.06% efficiency. *Nat. Energy* **9**, 592-601 (2024).

14 Guo, X. *et al.* Stabilizing efficient wide-bandgap perovskite in perovskite-organic tandem solar cells. *Joule*, doi:10.1016/j.joule.2024.06.009 (2024).”

Reviewer #2 (Remarks to the Author):

In this manuscript, the authors report a PEDOT:F-based interconnector layer (ICL) structure designed to minimize optical parasitic absorption, thereby enhancing the performance of n-i-p perovskite-organic tandem solar cells (P-O-TSCs). The study shows that optical losses associated with Au-embedded ICLs can be significantly reduced by tailoring the shape and size distribution of Au nanoparticles through manipulation of the surface properties of the underlying layer. The PEDOT:F-based ICL exhibits low surface free energy, promoting the formation of smaller and more homogeneous spherical Au nanoparticles, which effectively mitigate optical losses while preserving efficient recombination pathways. The optimized P-O-TSCs achieved a significantly higher current density, resulting in a champion efficiency of 25.34%. This work provides valuable guidance for the design of high-performance P-O-TSCs with minimized optical losses. However, several revisions are necessary before publication.

1. The authors have demonstrated that the thermal-evaporation deposition rates of Au significantly impact the optical properties of MoO_x and PEDOT:F-ICLs. This suggests that deposition rates critically affect the morphology of Au. Therefore, it is recommended to include SEM images as a function of deposition rates to better interpret these phenomena. Additionally, a comparison of the electrical properties of ICLs as a function of deposition rates should be included.

Response: We thank the reviewer's valuable suggestion. Indeed, the morphological characterization of Au and the electrical characterization of ICLs are critical for

understanding the effects of different Au deposition rates on the properties of ICLs. We have included SEM images of Au and J - V characterizations of ICLs in the revised SI and provided a discussion in the main text. It was observed that a speedy deposition rate primarily results in the reduction in Au nanoparticle layer quality, negatively impacting the transmittance of the ICLs. However, varying deposition rates of metal Au have negligible effects on the electrical properties of the ICLs, indicating that nominal 1-nm Au nanoparticles, as effective recombination sites within the ICLs, can effectively facilitate the recombination of electrons and holes, regardless of their morphology.

SI_Page 13, Figure S12.

“

Figure S12. (A) SEM images of 1-nm Au NPs deposited on MoO_x and PEDOT:F at different evaporation rates (0.02 Å/s, 0.20 Å/s, and 1.00 Å/s). Here, all samples for SEM measurement were based on ITO/MoO_x or PEDOT:F/1-nm Au structure. (B) The

corresponding J - V curves of ICLs (ITO/MoO_x or PEDOT:F/Au/ZnO/PFN-Br/Ag) with different deposition rates of Au (0.02 Å/s, 0.20 Å/s, and 1.00 Å/s).”

Page 10, line 235 of the text.

“As shown in Figure 2E, with the deposition rate increasing, the transmittance of MoO_x-based stacks at 500-930 nm improves, while it decreases significantly at longer wavelengths. Their overall transmittance curve tends to become lower and flatter, probably owing to the more dispersed shape distribution of larger Au NPs (Figure S12A). For PEDOT:F-based stacks, there is negligible variation in transmission for deposition rates of 0.02 Å/s and 0.20 Å/s (Figure 2F). However, at a speedy evaporation rate of 1.00 Å/s, minimal transparency forms in the visible and NIR regions. Obviously, this can be attributed to the reduction in nanoparticle layer quality under rapid deposition (Figure S12A). Nevertheless, the deposition rates of Au barely affect the conductivity of the ICLs (Figure S12B). Overall, the optimized PEDOT:F/Au combination consistently exhibits higher transparency.”

2. Hysteresis phenomena can significantly limit the stability of perovskite solar cells (SCs). Single-junction perovskite SC subcells exhibit pronounced hysteresis characteristics. Please provide the JV curves under forward and reverse scans for the various P-O-TSCs (e.g., D18-Cl:L8-BO:BTP-eC9 and PM6:L8-BO:BTP-eC9-based). Additionally, the stable power output (SPO) under maximum power point (MPP) conditions should be provided (e.g., 10 minutes).

Response: We appreciate the reviewer for drawing attention to the hysteresis phenomena and stabilized power output of tandem devices. We fully agree that the hysteresis behavior of solar cells is crucial for evaluating device performance and stability. P-O-TSCs in this work exhibit obvious hysteresis behavior between reverse and forward scans, which is significantly related to the pronounced hysteresis observed in CsPbI₂Br perovskite sub-cell. Regarding the stabilized power output (SPO), all P-O-TSCs achieve stable PCE with negligible performance degradation after 1000 seconds of maximum power point (MPP) tracking under solar simulator in air. As suggested, we have included the relevant data in the revised SI and provided a discussion in the main text, as shown below:

SI_Page 33, Figure S43.

“

Figure S43. *J*-*V* curves of (A) champion CsPbI₂Br-D18-Cl:L8-BO:BTP-eC9 TSC and (B) champion CsPbI₂Br-PM6:L8-BO:BTP-eC9 TSC under reverse and forward scans.”

SI_Page 33, Table S9 and Table S11.

“

Table S9. Device performance of champion CsPbI₂Br-D18-Cl:L8-BO:BTP-eC9 TSC with the PEDOT:F-based ICLs under reverse and forward scans.

Scan Direction	V_{oc} (V)	J_{sc} (mA/cm ²)	FF (%)	PCE (%)
Reverse	2.29	14.69	72.52	24.39
Forward	2.24	14.70	65.56	21.65

Table S11. Device performance of champion CsPbI₂Br-PM6:L8-BO:BTP-eC9 TSC with the PEDOT:F-based ICLs under reverse and forward scans.

Scan Direction	V_{oc} (V)	J_{sc} (mA/cm ²)	FF (%)	PCE (%)
Reverse	2.23	14.98	75.75	25.34
Forward	2.18	14.97	73.40	23.88

”

SI_Page 24, Figure S27.

“

Figure S27. Corresponding MPP tracking of champion CsPbI₂Br/D18-Cl:L8-BO TSC under solar simulator in air.”

SI_Page 34, Figure S44.

“

Figure S44. Corresponding maximum power point (MPP) tracking of champion CsPbI₂Br/D18-Cl:L8-BO:BTP-eC9 and CsPbI₂Br/PM6:L8-BO:BTP-eC9 TSCs under solar simulator in air.”

Page 16, line 352 of the text.

“Ultimately, the best CsPbI₂Br-D18-Cl:L8-BO TSC achieves a notable PCE of 21.12% (a stable power output of 20.5%, as shown in Figure S27), whereas the TSC utilizing MoO_x/Au only shows a PCE of 19.69%.”

Page 19, line 416 of the text.

“The device performance of champion CsPbI₂Br-D18-Cl:L8-BO:BTP-eC9 TSC under reverse and forward scans is summarized in Table S9 and Figure S43A. For the

stabilized power output under maximum power point (MPP) conditions, a stable PCE of ~23.72% is achieved with a negligible performance degradation (Figure S44).”

Page 19, line 426 of the text.

“The corresponding performance under reverse and forward scans is shown in Figure S43B and Table S11. Meanwhile, a stable power output of ~24.12% is monitored under MPP conditions after 1000 s (Figure S44).”

3. The morphology of Au is critical in determining the optical and electrical performance of ICLs. It is suggested that the authors replace the unclear SEM images of Au nanoparticles, such as those shown in Figure S9 (MoO_x) and Figures S10 (Batch 1 and 2), with clearer images.

Response: We sincerely thank the reviewer for your valuable comment. As suggested, we repeated the SEM measurements to obtain clearer images of the Au nanoparticles. The previously unclear images have now been replaced with clearer ones in Figure S8 (Batch 1 and Batch 2). However, as shown in the following figures, achieving clear images of Au nanoparticles on MoO_x at 1,000,000x magnification remains challenging due to the rapid degradation of the samples under the ultra-high electron beam. Additionally, the SEM instrument's maximum magnification (1,000,000x) limits the resolution of high-definition images. Nevertheless, we have made every effort to capture relatively clear images to support our findings, as shown in Figures S8 and S9. All new images have been added to the revised SI.

SI_Page 11, Figure S8 and Figure S9.

“

Figure S8. SEM images of 1-nm Au NPs on the MoO_x and PEDOT:F, captured under various magnifications. Here, the samples for SEM measurement were based on ITO/D18-CI/MoO_x or PEDOT:F/1-nm Au structure. The deposition rate of Au NPs was 0.02 Å/s.

Figure S9. SEM images of 1-nm Au NPs on the MoO_x and PEDOT:F. The samples from Batch 1 and Batch 2 were based on ITO/MoO_x or PEDOT:F/1-nm Au. The samples from Batch 3 and Batch 4 were based on ITO/ETL/PVK/D18-Cl/MoO_x or PEDOT:F/1-nm Au. The deposition rate of Au NPs was 0.02 Å/s.”

4. The D18-Cl-based ICL demonstrates Schottky-type barrier contact for hole injection from the ITO electrode, while quasi-ohmic linear characteristics are observed when the D18-Cl hole transport layer (HTL) is omitted. Is this due to the deeper energy level of D18-Cl? Will a carrier injection barrier be formed in the single-junction perovskite SCs? The authors should provide necessary clarification and discussion on this point.

Response: We thank the reviewer for highlighting this point. Yes, there is a Schottky-type contact barrier existing between ITO (WF \approx 4.7 eV) and D18-Cl with a deep HOMO of \sim 5.49 eV (DOI: 10.1002/adfm.202102413). Therefore, as shown in Figures S15B and S15C, when a forward bias is applied across the ITO/D18-Cl stack, the holes are unable to be injected from ITO, owing to the significant energy barrier between the HOMO of D18-Cl and ITO. When a reverse bias is applied, the holes can smoothly transfer from D18-Cl to ITO. This is directly reflected in the *J-V* curve of ITO/D18-Cl/MoO_x/Ag, as shown in Figure S16. In the first quadrant, when ITO is the positive electrode, the device shows significantly suppressed current injection, while in the third quadrant, holes can be smoothly injected from Ag, resulting in a large linearly increasing current. After introducing a PEDOT:PSS layer between D18-Cl and ITO, this issue is resolved, and the *J-V* characteristics of ITO/PEDOT:PSS/D18-Cl/MoO_x/Ag exhibit linear quasi-ohmic contact behavior.

However, the deep HOMO of D18-Cl does not affect the performance of single-junction perovskite devices, which are based on an n-i-p architecture. For CsPbI₂Br perovskite layer, its conduction band (e.g. 5.78 eV from 10.1002/aenm.201902279) is considerably deeper than the HOMO of D18-Cl, eliminating any energy barrier for hole transfer. Consequently, photo-generated holes in CsPbI₂Br perovskite layer can be efficiently extracted and transferred to Ag electrode via D18-Cl HTL.

We now incorporate the related data into the revised SI and give a detailed discussion, as outlined below:

SI_Pages 15-16, Figure S15 and Figure S16.

Figure S15. (A) Energy level diagram of ITO and D18-Cl at zero bias. (B) Energy level diagram of ITO and D18-Cl under forward bias. (C) Energy level diagram of ITO and D18-Cl under reverse bias.

Figure S16. Dark J - V curves of the devices with different architectures: ITO/D18-Cl/MoO_x/Ag, ITO/PEDOT:PSS/D18-Cl/MoO_x/Ag, and ITO/PEDOT:PSS/D18-Cl/PEDOT:F/Ag. First quadrant: hole injection from ITO.

Energy barrier between ITO and D18-Cl

Figure S15A shows the energy level diagram between ITO and D18-Cl at zero bias. It is evident that D18-Cl film exhibits a deeper HOMO than the work function of ITO. As illustrated in Figures S15B and S15C, applying a forward bias across the ITO/D18-Cl stack results in an inability to inject holes from ITO due to a substantial energy barrier between ITO and the HOMO of D18-Cl. Conversely, with reverse bias, holes can be efficiently transferred from D18-Cl to ITO.

This behavior is reflected evidently in the J - V curve of the ITO/D18-Cl/MoO_x/Ag device, as shown in Figure S16. When ITO is the positive electrode (first quadrant), current injection is notably suppressed. In contrast, with Ag as the positive electrode (third quadrant), holes are effectively injected, leading to a significant linearly increasing current. Introducing a PEDOT:PSS layer between D18-Cl and ITO can mitigate this issue, resulting in J - V characteristics for ITO/PEDOT:PSS/D18-Cl/MoO_x/Ag that display linear, quasi-ohmic contact behavior. However, the deep HOMO of D18-Cl does not impact the performance of single-junction perovskite devices with an n-i-p architecture. For CsPbI₂Br perovskite, the conduction band (e.g. 5.78 eV) is considerably deeper than the HOMO of D18-Cl, eliminating any energy barrier for hole transfer. Consequently, photo-generated holes in CsPbI₂Br perovskite layer are efficiently extracted and transferred to Ag electrode via D18-Cl HTL.”

5. By tailoring the composition of the organic active layer, a clear improvement in current density was observed. The authors should provide the external quantum efficiency (EQE) response of the optimized single-junction organic solar cells (OSCs), such as D18-Cl:L8-BO:BTP-eC9 and PM6:L8-BO:BTP-eC9, along with a labeled bandgap in the manuscript. Additionally, it is suggested to include a full spectrum (e.g., 300-1000 nm) EQE response in Figure S31.

Response: We appreciate the reviewer for this significant comment. As suggested, we have supplemented the EQE curves for the single-junction OSCs (without the anti-reflection film) in the revised SI, along with the labeled bandgap. Meanwhile, the full

spectrum (300-1000 nm) EQE response of tandem devices is also included in Figure S45, as shown below:

SI_Page 32, Figure S42.

“

Figure S42. Device performance and the corresponding EQE spectra of single-junction (A) D18-Cl:L8-BO:BTP-eC9 and (B) PM6:L8-BO:BTP-eC9 OSCs.”

SI_Page 34, Figure S45.

“

Figure S45. Corresponding EQE spectra of the champion P-O-TSCs utilizing PEDOT:F, incorporating D18-Cl:L8-BO:BTP-eC9 and PM6:L8-BO:BTP-eC9 organic rear cells. Reflection (denoted as 1-R) and sum (total EQE of individual sub-cells) curves are also presented.”

These revisions will enhance the clarity and comprehensiveness of the manuscript, providing a more robust understanding of the findings and their implications.

Reviewer #3 (Remarks to the Author):

In the manuscript titled "Overcoming Optical Losses in Thin Metal-Based Recombination Layers for Efficient n-i-p Perovskite-Organic Tandem Solar Cells," the authors address a crucial topic for tandem solar cells. The impact of thin metal-based recombination layers on optical losses is well-known, and the current consensus in the tandem community is that these losses are inevitable. Consequently, researchers have been focused on developing novel recombination layer structures to replace these thin metal nanoparticles. However, this work reports methods to minimize or eliminate these losses, offering new insights for advancing the field. While the device performances are impressive, some details in the manuscript are unclear and rarely discussed. Therefore, I recommend accepting the manuscript once the authors address the following issues to improve its quality further.

1. From the AFM measurements in Figure S4, the PEDOT:F layer exhibited a quite rough surface (RMS=6.17 nm) on top of D18-Cl. Usually, PEDOTs can form super flat films. Therefore, I am wondering whether this roughness comes from D18-Cl or insufficient interface contact. To make better understanding, the surface roughness of pure D18-Cl layer and pure PEDOT:F layer should be measured. Why does such high roughness of PEDOT:F have no obvious adverse effect on device performance?

Response: Many thanks for the reviewer’s comment. Yes, as evidenced by our experimental results in Figure S3, the PEDOT:F layer exhibits significant roughness on ITO/D18-Cl substrates. To eliminate potential artifacts, we re-evaluated the AFM

images for each layer on an ultra-smooth silicon wafer substrate, as shown in Figure S33 (below). The pure D18-Cl layer demonstrates a relatively low roughness of 2.36 nm, while the MoO_x layer exhibits negligible roughness. In contrast, the PEDOT:F layer shows substantial roughness, measuring ~5.42 nm. These findings confirm that the pronounced roughness of PEDOT:F arises from its intrinsic material properties, rather than the underlying D18-Cl layer. We hypothesize that the highly rough PEDOT:F layer forms incomplete layer-to-layer contact with D18-Cl, potentially creating microscopic voids or gaps at the interface, which adversely impacts the performance of P-O-TSCs, contrary to the assumption that roughness would have no significant effect. In our analysis of FF losses in PEDOT:F-based P-O-TSCs (as discussed in the main text, Page 16), we attribute the reduced FF (72.53%) to excessive series resistance in the complete ICLs utilizing D18-Cl/PEDOT:F/Au/ZnO/PFN-Br, largely due to the additional resistance introduced by the imperfect D18-Cl/PEDOT interface contact/morphology. Hence, for tandem devices, the roughness and uniformity of each layer within the ICLs are critical factors in achieving optimal performance.

Figure S33. AFM images of silicon wafer (Si), Si/D18-Cl, Si/MoO_x, Si/PEDOT:F, Si/D18-Cl/MoO_x and Si/D18-Cl/PEDOT:F.

2. To avoid misleading, the labels of Fig. S7 should be provided.

Response: We thank the reviewer for highlighting this point. In light of your suggestion, we now include the labels of the data in Figure S6 in the revised SI:

SI_Page 10, Figure S6.

“

Figure S6. Aspect ratios of 1-nm Au NPs on MoO_x and PEDOT:F, which were extracted from SEM images in Figures 1D-E.”

3. In the optical simulation, CsPbI₂Br with a 1.9 eV bandgap shows a perfect match with D18-Cl:L8-BO and PM6:L8-BO:BTP-eC9, resulting in the highest current density and efficiency in n-i-p structured perovskite-organic tandem cells (P-O-TSCs). However, the champion P-O-TSCs currently available are all based on ~1.85 eV perovskites. It seems to be challenging to achieve such high efficiency in this work. Specific explanations should be provided.

Response: We thank the reviewer for highlighting this point. In our optical simulation, we selected 6 typical perovskite components (FAPbI₃, MAPbI₃, CsPbI₃, CsPbI₂Br, CsPbIBr₂, CsPbBr₃) and combined them with the rear cells (D18-Cl:L8-BO and PM6:L8-BO:BTP-eC9) to simulate the tandems' current-densities ($J_{sc,TSC}$), resulting in 12 different bandgap combinations. The results indicate that when CsPbI₂Br is combined with D18-Cl:L8-BO (or PM6:L8-BO:BTP-eC9), which has a bandgap of 1.46 eV (or 1.41 eV), the highest matching current is achieved among six perovskite combinations. Here, to avoid any potential misunderstanding, we have modified Figure 3 and Figure S20 to more intuitive scatter plots, removing the misleading fitting trend line that suggested CsPbI₂Br/D18-Cl:L8-BO and CsPbI₂Br/PM6:L8-BO:BTP-eC9 were

optimal among all possible combinations. The revised figures are provided in the main text and SI as follows:

Page 36, line 762 of the main text.

“

Figure 3. Efficiency limits of 2-terminal perovskite-organic tandem photovoltaics.”

SI_Page 19, Figure S20.

“

Figure S20. Current-density and efficiency limits of 2T TSCs based on high-performance rear cell (1.41 eV, PM6:L8-BO:BTP-eC9) with different front cell bandgaps.”

Until now, all top-performing P-O-TSCs are based on a 1.80-1.85 eV / 1.33-1.38 eV bandgap combination (e.g. 25.82% PCE in Nat. Energy 2024, 9, 592-601, 25.22% PCE in Nat. Energy 2024, 9, 411-421, 25.13% PCE in 10.1016/j.joule.2024.06.009), which offers a higher theoretical efficiency limit. While the 1.89 eV / 1.41 eV combination in this work does not have the highest theoretical limit, it is still capable

of surpassing current record efficiencies. The $V_{oc,TSC}$ and $J_{sc,TSC}$ have already reached their maximum potential, with the FF being the only remaining limitation. If the FF can be improved to above 76% (ideally >80%), the overall P-O-TSCs could achieve efficiencies exceeding 26% (even >27%). Through further optimization, we are confident that this FF issue can be addressed.

4. Considering that this work centers on the properties of Au-based interconnection layers, I am curious about the stability of pure ICLs. Can the authors present more measurements to evaluate the stability of pure ICLs?

Response: We appreciate the reviewer's suggestion. We fully agree that the stability of pure ICLs is crucial for tandem solar cells. To accurately assess ICL stability, we monitored changes in the two most critical properties over time: transmittance and conductivity. Given that ICLs are primarily affected by light and heat during operation, we selected two test conditions: 85°C in N₂ in the dark, and 85°C in N₂ under 1 sun illumination. As expected, the optimized PEDOT:F-based ICLs demonstrated significantly superior stability. In contrast, MoO_x-based ICLs showed a notable improvement in transmittance in the near-infrared region, with conductivity proving more sensitive to photothermal conditions compared to PEDOT:F-based ICLs. We have incorporated the ICL stability data into the revised SI and included a detailed discussion in the main text, as outlined below:

SI_Pages 29-30, Figure S37 and Figure S38.

“

Figure S37. Stability of total transmittance of Glass/MoO_x or PEDOT:F/with or without Au/ZnO/PFN-Br stacks under (A) 85°C & N₂ & dark condition, and (B) 85°C & N₂ & 1 sun condition.

Figure S38. Storage stability of ITO/MoO_x or PEDOT:F/Au/ZnO/PFN-Br devices under (A) 85°C & N₂ & dark condition, and (B) 85°C & N₂ & 1 sun condition.”

Page 17, line 387 of the text.

“The PEDOT:F-based ICLs retain superior transmittance and conductivity after long-term exposure to 1 sun illumination and elevated temperature@85°C for 750 hours, as exhibited in Figures S37-S38. Interestingly, the MoO_x-based ICLs exhibit improved light transmittance beyond 615 nm after 750 hours of photothermal exposure, likely due to external light and heat-induced changes in Au morphology, resulting in a blue shift in its LSPR peak. Their conductivity significantly decreases at 85°C in the dark, yet remains relatively stable at 85°C & illumination, possibly due to accelerated Au diffusion reaching ITO electrode, which could contribute to a partial ‘recovery’ effect on conductivity.”

5. As shown in Figures 4G and 4H, the authors further optimized the photon utilization of tandem cells by employing various organic rear cells with narrower bandgaps. However, only the EQE spectra of rear cells are provided. Since the EQE of tandem

solar cells can reflect the carrier balance between the sub-cells, EQE spectra of the front perovskite cells should also be provided. In addition, discussions about the carrier balance between the sub-cells should be given.

Response: We appreciate the reviewer for the significant comment. As suggested, we have supplemented the full spectrum (300-1000 nm) EQE response for the tandem devices in the revised SI. The EQE spectra in Figure S45 disclose that both tandems employing D18-Cl:L8-BO:BTP-eC9 and PM6:L8-BO:BTP-eC9 rear cells yield almost identical integrated J_{SC} s for perovskite front cells and organic rear cells, demonstrating the balanced current match between two sub-cells.

SI_Page 34, Figure S45.

“

Figure S45. Corresponding EQE spectra of the champion P-O-TSCs utilizing PEDOT:F, incorporating D18-Cl:L8-BO:BTP-eC9 and PM6:L8-BO:BTP-eC9 organic rear cells. Reflection (denoted as 1-R) and sum (total EQE of individual sub-cells) curves are also presented.”

6. The operational stability under Maximum Power Point (MPP) conditions is an essential factor in assessing the performance of solar cells. Please provide the relevant MPP stability data for the tandem devices and the single-junction sub-cells, and give detailed explanations.

Response: We thank the reviewer for highlighting this point. Operational stability under Maximum Power Point (MPP) condition is crucial for assessing the performance of

solar cells. Unfortunately, due to the limitation of our laboratory stability testing equipment, we are temporarily unable to implement long-term MPP tracking tests of tandem devices. However, in light of your suggestion, we have supplemented the long-term operational stability data of tandems with different ICLs at the short-circuit mode, under continuous metal-halide lamp (MHL) illumination with an intensity of 85 mW/cm^2 in N_2 at $45\text{-}50^\circ\text{C}$. As anticipated, the tandem devices utilizing PEDOT-based ICLs exhibited significantly enhanced stability (60% after 750 hours) compared to those employing MoO_x -based ICLs (33% after 750 hours). However, for both tandems, we noticed a significant burn-in type degradation during the first 500 hours and before the tandem cells could fully stabilize, which is primarily attributed to the degradation of CsPbI_2Br perovskite front cells. We now include these stability data and a detailed discussion in the revised manuscript and SI, as shown below:

SI_Page 31, Figure S40 and Figure S41.

“

Figure S40. Normalized long-term operational stability of CsPbI_2Br -D18-CI:L8-BO P-O-TSCs employing MoO_x and PEDOT:F at the short-circuit mode, under continuous metal-halide lamp (MHL) illumination with an intensity of 85 mW/cm^2 in N_2 at $45\text{-}50^\circ\text{C}$.

Figure S41. Normalized long-term operational stability of single-junction D18-CI:L8-BO OSC and CsPbI₂Br PSC at the short-circuit mode, under continuous metal-halide lamp (MHL) illumination with an intensity of 85 mW/cm² in N₂ at 45-50°C.”

Page 18, line 399 of the text.

“A substantial improvement of the light operational stability in PEDOT:F-based tandems is also achieved, under continuous metal-halide lamp (MHL) illumination with an intensity of 85 mW/cm² in a nitrogen-filled chamber at 45-50°C (Figure S40). Nevertheless, a significant and gradual burn-in type degradation at the first 500 hours exists in both tandems. By individually exploring the device operational stability of single-junction PSCs and OSCs (Figure S41), it is revealed that the CsPbI₂Br sub-cell is the main culprit for the fast degradation, which undergoes swift degradation within initial tens of hours.”

Page 24, line 538 of the text.

“The long-term operational stability of the devices was assessed at the short-circuit mode, under continuous metal-halide lamp (MHL) illumination with an intensity of 85 mW/cm² in a nitrogen-filled chamber, with the temperature maintained at 45-50°C. The MHL provided light in the wavelength range of 400 to 800 nm.”

REVIEWERS' COMMENTS

Reviewer #1 (Remarks to the Author):

The authors have conducted strong experiments and simulations to address key questions, particularly regarding Au coverage and FF loss analysis, which contribute to a better understanding of the metal-based interconnect layer in tandem devices. Although the study lacks systematic experiments on the relationship between surface energy and the properties of the interconnect layer, the current data are sufficient to support the observed performance improvements. Therefore, I recommend this manuscript for publication in Nature Communications. Additionally, considering the resolution of AFM and the atomic scale, I suggest using nanometers (nm) as the unit and retaining two decimal places for the RMS value in Figure S33.

Reviewer #2 (Remarks to the Author):

The revisions have significantly improved the clarity and depth of the analysis. However, I have an additional point regarding to the EQE response and optical properties that need attention before publication:

-It is essential to provide the EQE response for single-junction OSCs, as well as details on the optical properties of these films when integrated with an anti-reflection layer. This is crucial because the EQE response of the organic subcell in P-O-TSCs (Fig. S45) appears unusually high compared to that of the single-junction OSCs (Fig. S42), especially in the 700 to 1000 nm wavelength range. This discrepancy raises questions about the optical enhancements attributed to the anti-reflection layer and requires further clarification.

Reviewer #3 (Remarks to the Author):

The authors have satisfactorily addressed the concerns I previously raised.

REVIEWERS' COMMENTS

Reviewer #1 (Remarks to the Author):

The authors have conducted strong experiments and simulations to address key questions, particularly regarding Au coverage and FF loss analysis, which contribute to a better understanding of the metal-based interconnect layer in tandem devices. Although the study lacks systematic experiments on the relationship between surface energy and the properties of the interconnect layer, the current data are sufficient to support the observed performance improvements. Therefore, I recommend this manuscript for publication in Nature Communications. Additionally, considering the resolution of AFM and the atomic scale, I suggest using nanometers (nm) as the unit and retaining two decimal places for the RMS value in Figure S33.

Reviewer #2 (Remarks to the Author):

The revisions have significantly improved the clarity and depth of the analysis. However, I have an additional point regarding to the EQE response and optical properties that need attention before publication:

-It is essential to provide the EQE response for single-junction OSCs, as well as details on the optical properties of these films when integrated with an anti-reflection layer. This is crucial because the EQE response of the organic subcell in P-O-TSCs (Fig. S45) appears unusually high compared to that of the single-junction OSCs (Fig. S42), especially in the 700 to 1000 nm wavelength range. This discrepancy raises questions about the optical enhancements attributed to the anti-reflection layer and requires further clarification.

Reviewer #3 (Remarks to the Author):

The authors have satisfactorily addressed the concerns I previously raised.

Point-to-point response to the Reviewers' comments

Manuscript #: NCOMMS-24-39617A

We sincerely thank all reviewers for the valuable suggestions, which have enabled us to substantially improve the quality of the manuscript. Detailed responses to each of the reviewers' comments are provided below.

Reviewer #1 (Remarks to the Author):

The authors have conducted strong experiments and simulations to address key questions, particularly regarding Au coverage and FF loss analysis, which contribute to a better understanding of the metal-based interconnect layer in tandem devices. Although the study lacks systematic experiments on the relationship between surface energy and the properties of the interconnect layer, the current data are sufficient to support the observed performance improvements. Therefore, I recommend this manuscript for publication in Nature Communications. Additionally, considering the resolution of AFM and the atomic scale, I suggest using nanometers (nm) as the unit and retaining two decimal places for the RMS value in Figure S33.

Response: We thank the reviewer for this helpful suggestion and fully agree with the importance of this detail. To improve rigor in describing the AFM results, we consistently report RMS values in Figure S33 in nanometers (nm) to two decimal places.

SI_Page 27, Figure S33.

“

Figure S33. AFM images of silicon wafer (Si), Si/D18-Cl, Si/MoO_x, Si/PEDOT:F, Si/D18-Cl/MoO_x and Si/D18-Cl/PEDOT:F. The pure D18-Cl layer demonstrates a relatively low roughness of 2.36 nm, while the MoO_x layer exhibits negligible roughness. In contrast, the PEDOT:F layer shows substantial roughness, measuring 5.42 nm. These findings confirm that the pronounced roughness of PEDOT:F arises from its intrinsic material properties. We hypothesize that the highly rough PEDOT:F layer forms incomplete layer-to-layer contact with D18-Cl, potentially creating microscopic voids or gaps at the interface, which adversely impacts the performance of PEDOT:F-based P-O-TSCs.”

Reviewer #2 (Remarks to the Author):

The revisions have significantly improved the clarity and depth of the analysis. However, I have an additional point regarding to the EQE response and optical properties that need attention before publication:

-It is essential to provide the EQE response for single-junction OSCs, as well as details on the optical properties of these films when integrated with an anti-reflection layer. This is crucial because the EQE response of the organic subcell in P-O-TSCs (Fig. S45) appears unusually high compared to that of the single-junction OSCs (Fig. S42), especially in the 700 to 1000 nm wavelength range. This discrepancy raises questions

about the optical enhancements attributed to the anti-reflection layer and requires further clarification.

Response: We thank the reviewer for highlighting this detail and agree that the EQE response of the organic rear cell in the tandem device is notably higher than in the single-junction OSC, which could be questioned by readers. Although an anti-reflective film was applied to the tandem devices, the EQE discrepancy of 700-1000 nm between the two devices exceeds typical expectations. The main reasons for this significant difference in EQE are as follows: (1) the use of an anti-reflective film, (2) a structural difference between the single-junction OSCs and the organic rear cell in tandems, and (3) the suboptimal performance of the single-junction OSCs in previous manuscript, showing the lower current density.

Regarding the second point, a previously overlooked detail is that, in the tandem configuration, the structure of the organic rear cell is .../ZnO/PFN-Br/BHJ/2PACz-Cl/MoO_x/Ag, whereas in the single-junction device, we used ITO/ZnO/BHJ/2PACz-Cl/MoO_x/Ag. However, the ignored PFN-Br layer strongly influences the crystallization of the active layer, the interface contact between ETL and BHJ, and overall device performance.

Additionally, during our response to the previous review round, unforeseen factors temporarily prevented our lab from achieving optimal performance in single-junction devices. As a result, the short-circuit current density and fill factor were relatively low, with efficiencies of only 16.10% for D18-Cl:L8-BO:BTP-eC9 and 15.36% for PM6:L8-BO:BTP-eC9, which led to the lower EQE responses in the single-junction devices compared to optimal conditions.

To address this issue, we fabricated new devices with the updated structure, and compared their performance and EQE spectra with and without the anti-reflective film. Additionally, to further demonstrate the role of the anti-reflective film, we included its effect on the device's reflection spectrum (denoted as 1-R), as shown in the Figure S42 (below). The devices showed improved overall performance, with an increase in current density of ~ 0.65 mA/cm² after applying an anti-reflective film. In the EQE spectra from 700-1000 nm, the highest EQE response exceeded 85%, aligning well with the response of the organic subcell in the tandem devices.

“

Figure S42. Device performance and corresponding EQE spectra of single-junction (A) D18-Cl:L8-BO:BTP-eC9 and (B) PM6:L8-BO:BTP-eC9 OSCs with or without an anti-reflection film (ARF). Note that the device structure is ITO/ZnO/PFN-Br/BHJ/2PACz-Cl/MoO_x/Ag. Reflection (denoted as 1-R) curves of the devices are also presented.”

Reviewer #3 (Remarks to the Author):

The authors have satisfactorily addressed the concerns I previously raised.